# A Case Report on How BOAM Offers a Brief Family-Based Treatment by Integrating Psychoeducation and Self-Diagnostics

**DOI:** 10.3390/ijerph22040559

**Published:** 2025-04-03

**Authors:** Eva S. Potharst, Damiët Truijens, Francisca J. A. van Steensel, Steve Killick, Susan M. Bögels

**Affiliations:** 1UvA Minds, Academic Outpatient (Child and Adolescent) Treatment Centre of the University of Amsterdam, Banstraat 29, 1071 JW Amsterdam, The Netherlands; d.truijens@boam.eu; 2Research Institute of Child Development and Education, University of Amsterdam, Nieuwe Achtergracht 127, 1018 WS Amsterdam, The Netherlands; f.j.a.vansteensel@uva.nl; 3The George Ewart Evans Centre for Storytelling, University of South Wales, Cardiff, Wales CF24 2FN, UK; 4Developmental Psychology, University of Amsterdam, Nieuwe Achtergracht 129-B, 1018 WS Amsterdam, The Netherlands; s.m.bogels@uva.nl

**Keywords:** youth mental health, development, child psychopathology, executive functioning, parenting stress, diagnostic system, intervention, family functioning, self-regulation

## Abstract

BOAM is a family-based method in which children and parents together create an explanatory, personal and systemic diagnosis. Based on ten playful and visual models, the therapist provides universal psychoeducation to gain insight into the personal, relational and contextual causes of the child’s problems for a shared understanding of how to approach them. This case report describes a seven-session BOAM trajectory in a family with a 6-year-old child with emotional and behavioural dysregulation, such as frequent temper tantrums, hitting her infant sister, and threatening with knives. In this case report, the course of the sessions is described, including the way the family applied the BOAM models within their (cultural) family values. The mother completed questionnaires on child psychopathology (Child Behaviour Checklist), executive functioning (Behaviour Rating Inventory of Executive Function), parenting stress (Parenting Stress Index) and partner relationship (Family Functioning Questionnaire) at baseline, pretest, post-test, and 3- and 5-month follow-up, and the father completed questionnaires on child psychopathology and parenting stress at baseline and 5-month follow-up. Parents reported clinically significant improvements, as calculated with reliable change indexes, in child externalising psychopathology, self-regulation, and parenting stress (post-test and 3- and 5-month follow-up). BOAM is a short and accessible method for psychoeducation, diagnostics and treatment. BOAM seems to be an effective intervention for this family; however, more research is necessary to demonstrate its effectiveness. This case report painted a vivid picture of how family conversations can be structured and targeted using the models.

## 1. Introduction

Children’s self-regulation is foundational for healthy development and functioning and is predictive of later mental health and adaptation to the environment [1]. Self-regulation is described as the ability to control or direct attention, thoughts and emotions and to adjust behaviour to adapt to a given situation [2]. The development of self-regulation goes hand-in-hand with neural and cognitive development [3,4]. However, environmental factors—and during (early) childhood, especially parental factors—also play an essential role in the development of self-regulation [5,6,7], a process that is called co-regulation. Because of the important role that parents play in the development of self-regulation in children, parental involvement in interventions aimed at improving children’s self-regulation is essential [8]. A recent meta-analysis on the treatment of child symptomatology related to self-regulation showed that interventions involving parents were more effective than interventions that only involved the children [9]. Also, a systematic review of family therapy and systemic interventions showed that family-based interventions are effective for children’s emotional and behaviour problems [10]. Parental involvement is of utmost importance not only during the treatment phase of mental health care but also in the assessment phase [11,12] and in essential elements such as shared decision-making [13] and psychoeducation [14,15]. While there are many family-based manualised interventions, there are few equivalents of such programs for the diagnostic process. The current case report describes the application of an innovative family-based diagnostic method (BOAM [16,17]) in a family with a child having problems with the self-regulation of emotions and behaviour, which is introduced further in Section 2 (Detailed case Description).

### 1.1. A Family-Based Diagnostic Method: BOAM

BOAM is an acronym for the four core psychological needs of every human being in hierarchical order: basic needs, ordering, autonomy and meaning. Ordering is a new term for the unconscious process in which the psyche converts the constant stream of sensory stimuli into a personal perception of reality, in response to which the thoughts, feelings and behaviours arise. Fulfilling these four core needs is seen as the core task of the psyche in BOAM theory, which connects knowledge elements from psychology, neuropsychology, pedagogy and sociology in a relatively simple way. Based on this theory, children and their parents receive psychoeducation during the diagnostic process, which is also the start of the treatment. Psychoeducation means that the child and/or their families are offered information, advice, and coping strategies regarding their mental health and mental health problems, maintain factors and potential interventions [18]. This is performed using models, which are mindmaps with pictures with clear imagery and minimal text. The BOAM theory and models were developed by author DT while she was working with families with children with complex problems. In creating the models, she found inspiration in several psychological theories, models and treatments, such as Maslow’s hierarchy of needs model [19], Dawson and Guare’s executive functioning coaching model [20], schematherapy [21], cognitive behavioural therapy [22], and non-violent resistance in families [23]. The principles of the development of the BOAM method are explained in more detail after a short overview of other diagnostic methods.

### 1.2. Diagnostic Approaches

Generally, when a child is admitted to mental health care, a diagnostic process is the first step. This process often consists of interviews with different informants, questionnaires, tests, and observations, to make an inventory, description, ordering and categorisation of the problem behaviour, which then usually results in a classification of the problem. Often, a classification system such as the fifth version of the Diagnostic and Statistical Manual of Mental Disorders (DSM-5) is used for this [24]. The transdiagnostic approach identified several problems related to classifying problematic behaviour, such as that mental problems are not clearly separable from each other. Also, not only biological but also psychological and social factors play a role in the development and maintenance of mental health problems, which does not become clear with a classification. Furthermore, classification is often not enough to guide the treatment process or to offer answers about the cause of the problems, while a diagnostic process is also aimed at explaining the problems. In answer to some of these issues, several alternative diagnostic systems have been developed, such as the Hierarchical Taxonomy Of Psychopathology (HiTOP) [25] and the Psychodynamic Diagnostic Manual (PDM-2) [26]. Both are highly specialised and do not seem to be accessible enough to use in close collaboration between the child, parents and therapist.

### 1.3. The Guiding Principles in the Development of BOAM

The first guiding principle in the development of BOAM was the need to develop a family-based diagnostic method. In a BOAM trajectory, the child and the parents together receive universal psycho-education. Both parents and child have an active role in applying psychoeducation to the child’s development and problems so that an understanding of the cause of these problems emerges. Furthermore, parents are also invited to apply psycho-education to themselves, integrating insights on their role in the explanatory diagnosis of their child.

The second guiding principle was the equal collaboration between the therapist, the child and the parents, not only during the treatment phase but from the start of a mental health care trajectory. It supports an active engagement of children and parents, as well as the development of a positive working relationship between the therapist and the children and parents, which are predictive of treatment success [14,27,28]. Diagnostic assignments are performed by the parents (and, depending on the age and motivation of the child, also by the child), which leads to the explanatory diagnosis. Automatically, the experiences and perspectives of all family members are represented. This is important, as the feeling of not being listened to or dismissed is one of the most mentioned barriers for parents to seek psychological support for their child [29]. In the literature, some information can be found on collaborative diagnosis, in which the client is actively involved and feels valued [30]. It has been shown that clients find both the process of collaborative diagnosis and the resulting diagnosis more meaningful, informative and useful [30]. It has also been suggested that collaboration could mitigate some of the reported negative consequences of diagnosis, such as feeling stigmatised and disempowered [30]. Another approach that is interesting in this regard is therapeutic assessment, which is aimed at providing the client with therapeutic benefits from the assessment process itself and changing their narrative about themselves and their environment to a more positive one [31]. However, there is very little evidence for this approach [31].

Third is the principle that treatment should start as soon as possible. Regular psychological assessments involving psychological tests, clinical interviews, behaviour rating scales, self-report questionnaires, behavioural observations, projective–expressive assessment and multiple informants [32] before the start of the treatment phase may delay the treatment phase several weeks or even months. During the BOAM trajectory, parents receive universal psychoeducation together with their child using ten playfully illustrated BOAM models. They apply each model to their own situation, guided by the therapist, thus discovering for themselves the exonerating and explanatory system diagnosis for the child’s psychological problems. During their self-diagnostic process, the family can learn to mentalise experientially, develop shared insights and conversation skills, and come to shared decision-making about the parenting approach to the problems and how to involve their own network. In this way, there is no treatment delay. In the literature, self-diagnosis refers to a growing phenomenon that individuals, especially adolescents, are diagnosing themselves with a mental disorder as a way to understand themselves better [33]. These diagnoses usually take the form of a classification as described in the DSM-5. In BOAM theory, self-diagnosis refers to a descriptive and explanatory diagnosis that gives clarity on both the form of expression of the mental problems and the causes and interplay with the family and broader environment.

The next guiding principle to be mentioned is that a child’s problems should be viewed in its context. The ten visual models of the BOAM method can be applied not only to the target child but also to other family members, including the parents. The BOAM method does not only take the target child and its family into account but also the broader environment, such as school, social media and society, which is also important in child development [34,35]. Understanding the child in the context opens the possibility for change and growth and is therefore less static than classifying the child’s behaviour and emotional problems only.

The last guiding principle in the BOAM method that should be mentioned is that BOAM should be suitable for families with different cultural and ethnic backgrounds or children who identify as LGBTQIA+. One of the central concepts of BOAM is the psychological needs of children. These psychological needs can be fulfilled according to the norms and values of the family members, which stimulates their self-solving capacity in changing their parenting style and parenting choices. Implementing one’s own life goals, needs, norms and values makes families feel seen and understood and makes them motivated during treatment [36,37,38]. This is highly determinant of a positive treatment effect [39]. A side note is that some parenting norms and values might contradict children’s rights. With the help of the model depicting children’s basic needs, the therapist can make this a topic of conversation. In general, the visual models help in communication, which is especially beneficial if the mother language of the family and the therapist differ or if, for example, a child has a language delay. The fact that BOAM models depict clear imagery and minimal text makes psycho-education easier, faster and better understood [40].

### 1.4. A BOAM Trajectory for Families: The Current Case Report

Considering these guiding principles, the BOAM method is developed for families with children with any difficulties in emotional and behavioural regulation or mental health problems. The full name is ‘BOAM-self-diagnostics’, and the basic protocol consists of seven sessions with manualised additions for complex family problems. During the treatment, they themselves create an exonerating and explanatory systemic diagnosis, which is thus naturally supported by all involved. In this process, the family can experientially learn how to mentalise, develop insight and better conversational skills, tap into their own problem-solving abilities, and come to joint decision-making about how to address problems. In practice, careful diagnosis, psychoeducation and an immediate start of treatment coincide in this way, which is helpful for the family and relieves the healthcare system.

The family in this case report gains knowledge about the four core psychological needs (including ordering); the unconscious ordering processes; the role basic psychological needs and developmental stages play in them; and how self-regulation, practical and social skills, behavioural strategies and relationship dynamics emerge from them. This allows them to discover what role all this plays in their own lives and relationships and to understand themselves and each other better. The aim of this case report is (1) to illustrate a BOAM trajectory, including the use of the BOAM models in clinical practice in a child mental health setting, and (2) to study the effect of the BOAM trajectory on the participating family.

## 2. Detailed Case Description

The focus of this case report is on the 6-year-old girl, Victoria, with emotional and behavioural problems, and on her parents. The emotional and behavioural problems that were present were daily recurring anger, temper tantrums, aggression towards family members and tics. Their case was chosen to illustrate how the BOAM method can be used in collaboration with parents and children, even if they are still young, and to illustrate how the BOAM method integrates family-based diagnostics, psychoeducation and treatment, thereby making it unnecessary to give children individual treatment. This case was also chosen because it is a representative example of how the focus on the development of greater self-insight and empathy in parents, based on universal psychoeducation, can activate their self-solving abilities and strengthen parenting skills. Furthermore, there is a need for interventions for children that are personalised and that consider the unique perspectives and needs of neurodiverse children [41], which is what this girl was shown to be later on in the trajectory. Lastly, this case describes a BOAM trajectory of regular length. Our first pilot study on the BOAM method focused on families with children who were non-respondent to regular mental health care and often needed a longer trajectory [17]. The family, in this case report, had received no previous mental health care.

This study was approved by the ethical review board of the University of Amsterdam (2017-CDE-8422 and FMG-11997-2024). The parents provided written informed consent when they were included in the study and again after reading this case report. Part of the questionnaires that were used in this study were derived from the pretest and post-test of the general effectiveness study of the mental health care centre that the girl was admitted to. Both parents completed these questionnaires before the intake and when the file was closed. In the current case report, these two measurement occasions are called the waitlist assessment and the 5-month follow-up assessment. Furthermore, both parents were asked to complete questionnaires that were part of the BOAM study. At the time, the father was too busy for this, but the mother completed these questionnaires at three time points (in the current study called the pretest, post-test and 3-month follow-up assessments). More information on the questionnaires that were administered is included in Section 2.4 Measures, which follows the description of the BOAM trajectory.

### 2.1. Family Intake Information

The intake was performed in week 1 in a child mental health centre located in a large city in the Netherlands by a child psychologist. The family consists of a 34-year-old mother; a 40-year-old father; a 10-year-old son, Alexander; the 6-year-old, Victoria; and a nearly 1-year-old daughter, Anastasia, called ‘the infant’ henceforth. The parents came from an eastern European country and moved to the Netherlands before they had children. The mother has her own business and works from home, which she combines with the care of the children. The father works full-time and works abroad for several days a week. Victoria and Alexander go to a sociocratic school without classes or a fixed learning schedule; children follow their own interests, and social rules are established in a group process. Victoria leans heavily on Alexander when it comes to socialising at home, in school and in scouting; she wants his friends to also play with her, or she wants him to play with her. The infant is at home and does not go to daycare. The parents have prioritised a secure attachment relationship with their children by providing a lot of physical closeness (e.g., by sleeping together) and being present for them as much as possible. Furthermore, they find it important to parent their children without punishing or rewarding them.

### 2.2. Child Intake Information

Victoria was admitted to a child mental health care centre because of behavioural difficulties. Since the age of 2, she has shown a lot of anger, especially when things go differently than she expects or wants. Every step of the daily routine (e.g., dressing, eating, etc.) is a struggle and a source of conflict. She has tantrums several times per week to several times per day, tends to express sadness as anger and stays angry for a long time. Victoria shows a lot of controlling behaviour, for example, by demanding the exact way her mother should hold her hand, the pace at which she should walk (not too slow and not too fast) and the way she should braid her hair (not too high and not too low). If such conditions are not met, she does not cooperate and, for example, will not go to school. If she thinks her mother did something wrong, she requests that she say sorry several times.

When she gets angry, she tends to hurt Alexander, Anastasia, and her mother, after which she does not seem to show any feelings of guilt. At the age of 5, she bit her mother so hard that her mother still has scars. Victoria tends to threaten when she becomes angry or when she wants something, such as to stab or cut someone while actually holding a knife. In the car, Alexander tends to hold Victoria’s hands to keep her from hitting him or the infant. Also, in contact with other children, Victoria tends to hit or bite when things do not go her way, causing her parents to avoid inviting other children into their home.

When Victoria was almost 2 years old, her mother brought her to a nanny when she had to work. She resisted this fiercely by kicking, screaming and not allowing her mother to dress her. As a toddler, she preferred certain sets of clothes, refusing to wear other clothes if the preferred clothes were washed. She could not wait a while and currently still demands an immediate response.

When Victoria suffers from stress, she shows tics (squinting her eyes and blinking one eye a lot). When she is not angry, her mood is usually neutral to positive, and in those moments, she is kind to the infant. Victoria’s hobby is dancing, which she is good at. She says she also likes to play and swing, although she finds it hard to play by herself.

During the intake, Victoria was shy around the psychologist. She made eye contact and reacted non-verbally. The infant was also present at the intake; she played and sat with the mother. During the intake, Victoria sometimes went to the mother to give her a hug.

The parents feel strongly about the parenting style they employ, and they requested a therapist who would respect the choices they make in parenting. They also did not want Victoria to receive a diagnostic classification or individual therapy. Because of this wish, a BOAM trajectory was indicated instead of regular assessment at the mental health care centre followed by treatment. The BOAM trajectory is not offered by the intake psychologist but by another therapist (DT). Meanwhile, the child psychologist who conducted the intake stayed available for anything that might be additionally required during the BOAM trajectory. The BOAM trajectory started after three weeks and consisted of an introductory session, seven sessions on a 2-weekly basis, an evaluation session, a follow-up session, and an evaluation with the intake psychologist.

### 2.3. BOAM Trajectory

#### 2.3.1. Introductory Session: Meeting the BOAM Therapist (Both Parents, Week 4)

The goal of this meeting was for the parents to become acquainted with the BOAM therapist and to answer the question of whether the parents indeed wanted to start with the BOAM trajectory together with Victoria. The mother talked about the problems that she has encountered with Victoria. She said Alexander protests the unequal treatment he receives compared to his sister. She hopes to obtain tools to reduce his protest and seems to unconsciously assume that the large treatment difference cannot be reduced because it is impossible to improve Victoria’s functioning. Alexander seems to receive and take on much responsibility regarding the monitoring and supervision of his sister and co-regulating her emotions. This is not only at home but also at school where they are in the same group. The therapist argued that this seems to play a role in the fact that relatively few problems arise in school. The mother agreed with this assumption and said she was happy about that. The therapist expressed her concern about the burden Alexander is carrying and the impact it may have on him, which seems to be a new perspective for the parents. The therapist also enquired if they have worries about how the behaviour problems may develop in the future, which they said they have not thought about yet. The therapist explained that when they can find the causes for Victoria’s behaviour with the help of BOAM self-diagnostics, it might become possible to change her behaviour through parenting that is more adjusted to her specific needs. When the parents spoke about the problems, their tone and facial expressions remained flat. After the therapist raised the possibility of positively adjusting Victoria’s development, the parents seemed to be more involved and motivated for the BOAM trajectory. The parents indicated that Victoria is not keen on the sessions, and they want to stress her as little as possible. The therapist and parents agreed on the BOAM trajectory and confirmed that part of the sessions would be performed with them as parents and part of the sessions would be performed with Victoria present.

#### 2.3.2. BOAM Session 1: Model 1 (The Whole Family, Week 6)

The parents brought the whole family to the session; they were used to doing activities with the family as a whole and did not realise that the presence of the combination of the three children would distract them from the session. When meeting the therapist, Alexander and Victoria both acted shy and polite. Alexander started playing and sometimes took care of the infant. Victoria did not seem to know what to do. She kept walking to her mother or father, did not answer any questions from the therapist, and looked at her suspiciously. The mother stated aloud that it is important for all of them to accept that Victoria is ‘ill’ and that this illness cannot be cured. She says she hopes that when this is clear, Alexander and the outside world will stop keeping her responsible for the behaviour of Victoria. The mother explained that other parents—who never found out what was wrong with their child—told her that this had made their burden heavier.

At a certain point, the father started playing with Victoria. Meanwhile, the therapist repeated to the mother that when they are able to find the causes for Victoria’s behaviour, it can become possible to change it by adjusting parenting better to her needs. The therapist gave psychoeducation based on Model 1 (the BOAM developmental model, see Figure 1) and the explanation of Model 1 (Box 1).

The most important understanding that comes from this model is that psychological and/or behavioural problems develop if *nurture* is not adequately attuned to a child’s *nature*. This understanding changed the mother’s perspective; she realised that they, as parents, may be able to help change Victoria’s behaviour. This stresses the importance of her finding explanations for her behaviour. The therapist expressed some concern about the term ‘illness’ that the mother uses in the presence of Victoria and Alexander. They agreed that only the parents (or the parents and the infant) would come for the next appointment.

**Figure 1 ijerph-22-00559-f001:**
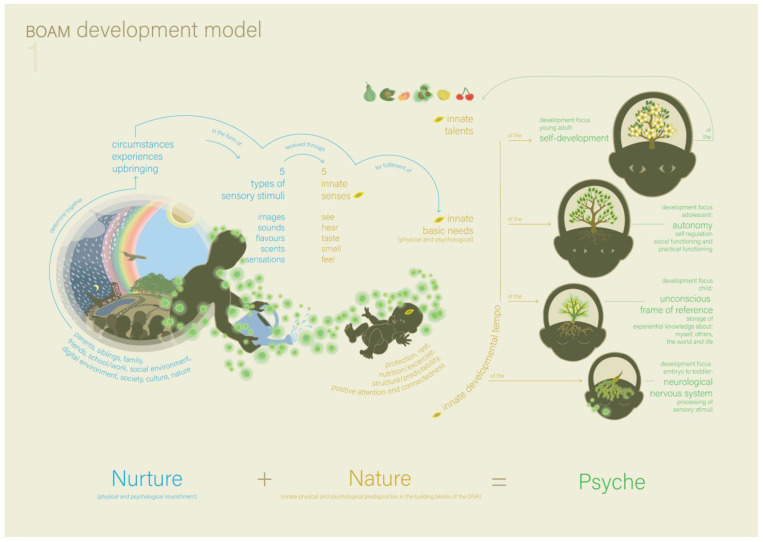
The BOAM Development Model (Model 1) [16].

Box 1Model 1: the BOAM Developmental Model (Figure 1).Model 1 shows the development of a human psyche that is represented as a growing tree. The tree seed represents the nature of the psyche (innate senses, basic needs, talents and pace of development), and the soil in which it lands represents the nurture of the psyche (parents, family, school, social and digital environment, society, culture). All (positive and negative) nurture consists of experiences composed of sensory stimuli entering the senses. This means that nobody perceives reality directly; instead, our neurological and psychological processes determine how we experience reality. The better nurture is aligned with nature while growing up, the more accurate the unconscious frame of reference (about the self, others, the world and life) can develop, and the better the psychological development can be. Within the tree, the root system represents the neurological nervous system, and the branch system represents the unconscious frame of reference.

#### 2.3.3. BOAM Session 2: Model 2 and 3 (Both Parents and the Infant, Week 8)

Model 2 (the BOAM basic model, see Figure 2, and the explanation in Box 2) was explained to the parents. The parents understand the model immediately and also understand that Victoria’s need for order is not fulfilled, which means that she has ‘ordering problems’ that lead to a dysfunctional perception of reality. Both parents recognised the characteristics of ordering problems in their daughter; her need for structure and predictability is much stronger than Alexander’s at the same age. They recognise that Victoria finds it difficult to oversee and understand situations, which undermines her basic security. The father added that, from this perspective, it makes sense that changes always lead to anger and resistance. Model 3 (the BOAM autonomy model, not included in this article, and the explanation in Box 3) is presented and discussed shortly. From this model, the parents understood that when Victoria’s need for ordering is not fulfilled (for example, because of sensory overload or lack of clarity), she loses her basic safety, is incapable of functional behavioural strategies and loses her self-regulation.

Box 2Model 2: the BOAM Basic Model (Figure 2).Model 2 shows a tree that represents the human psyche and is based on the Pyramid of Maslow. It explains that the human psyche is constantly focused on the fulfilment of four core psychological needs (in hierarchical order: basic needs, order, autonomy, and meaning), of which the starting letters form the acronym BOAM. The basic needs of the psyche (represented by the soil the tree grows in) are protection, rest, nutrition/exercise, structure/predictability, positive attention and connectedness, all of which need to be fulfilled both physically and psychologically to be able to function well. If the basic needs are fulfilled, the next need to be fulfilled is the need for ordering. Ordering is the neuropsychological process by which the continuous flow of sensory stimuli is converted into a perception of reality. This is performed quickly and unconsciously by using the unconscious frame of reference about the self, others, the world and life, which is built up over a lifetime (depicted by the tree trunk and branches). The frame of reference contains not only core beliefs and cultural values but also foundational knowledge about practical and social structures. Because the frame of reference largely determines the perception of reality, it also determines the thoughts and feelings that are a reaction to that perception and thus cause the behaviour. This behaviour influences the new experiential knowledge that is collected, and in this way, this frame of reference is further refined. The more comprehensive the frame of reference is, the faster and better the ordering processes can take place, and new experiences can be understood. When the need for ordering is fulfilled, basic security can be experienced, and the subsequent need for autonomy (depicted by the leaves) arises. Only when the (imperceptible) ordering goes well enough (meaning the perception of reality sufficiently matches (other‘s) reality) can behavioural strategies emerge that lead to (perceptible) autonomy. The core need for autonomy is fulfilled by self-regulating behaviour, such as executive functioning and social abilities, which allow self-organisation of basic needs in practical and social circumstances. After fulfilment of the need for autonomy, the need for meaning arises, which consists of different forms of self-development (depicted by blossom) and, in the case of adults, servitude (depicted by fruits or nuts).

Box 3Model 3: the BOAM Autonomy Model (not included in this article).Model 3 zooms in on the middle (two) layers of the BOAM basic model and shows that behaviour strategy arises unconsciously from the perception of reality. New situations are ordered quickly, automatically and unconsciously with the help of the unconscious frame of reference about the self, others, the world and life. This frame of reference is always subjective because it arises from a person‘s unique nature and unique nurture. Therefore, the reality perception that is based on this frame of reference constitutes one‘s personal idea of cause-and-effect in the current circumstances and may be more or less coherent and adequate. A more functional reality perception leads to a more functional behaviour strategy. Depending on the functionality of the behaviour strategy, self-regulation can occur: thoughts, feelings and behaviour can be efficiently directed towards the fulfilment of the basic needs or need for Meaning. Functional behaviour consists of 12 core psychological functions including, for example, concentration, taking initiative, flexibility and bounding. These functions can be deployed both practically and socially.

Both parents had already assumed that Victoria was ‘unable’ and not ‘unwilling’, and after discussing the models, they seemed relieved that these models confirmed this assumption. The parents had translated this understanding into their parenting practices by not punishing Victoria and not becoming angry with her. The therapist complimented the parents for this, which seemed important for them. They visibly relaxed more. The mother shared more about the difficulties that arise (not being able to work when her daughter is present and her own fatigue). The father has fewer obligations when he is at home with the children and copes by playing with Victoria a lot and by taking her out for something else to eat when she does not want what has been cooked. Because the first sessions were still focused on gaining insights, new ways of dealing with Victoria’s behaviour were not discussed. The main hypothesis arising from this session is that she suffers from ‘ordering problems’ that negatively influence her self-regulation.

In the next session, Victoria would be present again. The therapist explained to the parents that she could play or draw during the sessions but not read or use digital devices so she could hear what was being discussed and may choose to react. The homework assignment was for the parents to analyse difficult situations with the help of Models 1 to 3.

#### 2.3.4. BOAM Session 3: Models 4 and 5 (Both Parents, Victoria and the Infant, Week 10)

At the beginning of the session, Victoria was drawing. The therapist only asked some light-hearted and ‘closed’ short questions, to which she nodded yes or shook no. Victoria did talk to her parents in their own language. If the therapist showed, in a funny way, that she did not understand what was being said and told them that she was very curious about what they said, Victoria laughed and looked triumphant. The therapist reassured Victoria that she would only talk with the parents today, after which Victoria seemed to relax.

The parents told the therapist that, based on the hypothesis they formed in the last session, they tried to make situations more structured and predictable for Victoria. They saw a positive effect; Victoria seemed to listen better to her parents. The parents and therapist applied the models to a few recent situations in which difficulties occurred in order to analyse and better understand the cause of these difficulties. The new models they used were Model 4 (the BOAM Disturbance Model, see Figure 3, and the explanation in Box 4), and Model 5 (the BOAM Effect Model, see Figure 4 and the explanation in Box 5).

**Figure 3 ijerph-22-00559-f003:**
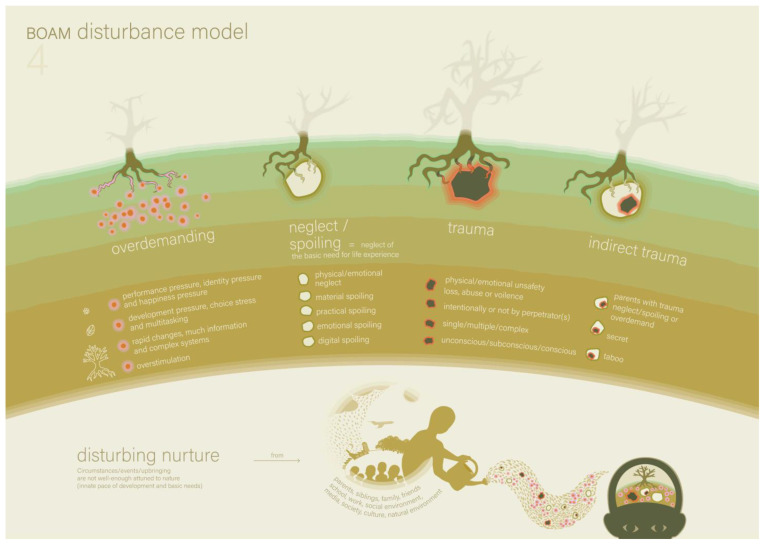
The BOAM Disturbance Model (Model 4) [16].

Box 4The BOAM Disturbance Model (Figure 3).Model 4 shows the four different categories of circumstances in which *nurture* is not adequately attuned to a child’s *nature*, which disrupts basic safety and the development of a functional frame of reference. The categories are depicted as different distortions in the ground in which the tree grows, such as too much fertiliser, stones and holes in the soil. Category 1 is called overdemanding and is quite common in the 21st century. The experiences of a child are not negative per se, but (1) the experiences can be too overstimulating for the nervous system, (2) there may be too many experiences in a (too) short amount of time for the ordering processes, and/or (3) there can be too much pressure on development and performance. Category 2 is called neglect, which can be physical and/or emotional and relates to one or more basic needs. A special form of neglect occurs in the case of spoiling (material, practical, emotional or digital), with the consequence that the basic need for life experience is insufficiently fulfilled. Spoiling is often an intuitive parental response to an overdemanding child, with the intention of protecting them from too many psychological failure experiences. Category 3 is called trauma, which can be physical and/or emotional, accidental or through intentional perpetration. Trauma can have a singular or multiple-complex effect on the unconscious core beliefs and experiential knowledge in the frame of reference. Category 4 is called indirect trauma, which arises because of a secret, taboo or psychological problem in the parenting system. When dysfunctional adult behaviour is not directed at the child, no direct trauma occurs; however, indirect trauma can occur due to a lack of ‘emotional predictability’.

**Figure 4 ijerph-22-00559-f004:**
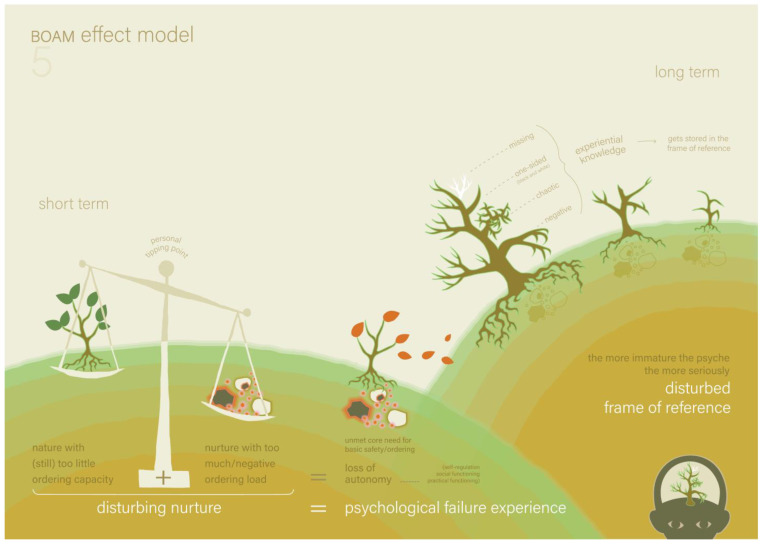
The BOAM Effect Model (Model 5) [16].

Box 5The BOAM Effect Model (Figure 4).Model 5 shows the short- and long-term consequences of all four categories of disturbances. The short-term consequence is depicted as an unbalanced weighing scale and a tree that loses its leaves as a result. This means that if the ‘ordering load’ of a certain situation (*nurture*) is too large, too negative or both, compared to the actual state of the nervous system and frame of reference (*nature*), the child noticeably loses autonomy. This means the loss of self-regulation and of the 12 core psychological functions (practically or socially), and this is called a psychological failure experience. Based on the BOAM basic model, this was caused by the unnoticeable loss of the ordering and basic safety. Depending on the ordering capacity, a person has a personal ‘tipping point’ at which circumstances are too overdemanding or too negative to process. The long-term consequence of too many or too severe psychological failure experiences causes the unconscious frame of reference not to develop properly. As a result, the required experiential knowledge in the frame of reference may become missing, one-sided (black and white), chaotic or negative.

One of the situations that were analysed was a dinner in which Victoria refused to eat what her mother cooked, even though it was a meal that she usually eats. After taking one bite, she said it did not taste good, became angry and refused to eat. Her father thought that she genuinely had trouble eating during those times. Because it is important that she eats enough, he takes her to the gas station, where she gets a sausage sandwich that she likes. The parents said that Victoria could be very sensitive to changes in her food: small changes in the taste, texture or colour can lead to her refusing to eat. The therapist pointed to ‘Overdemanding’ in Model 4 and said that this might be the cause of this behaviour. If a child is chronically over-stimulated neurologically, the sensations of eating can be overwhelming. The parents recognised that Victoria seemed to be overstimulated a lot and that any change in a situation, such as the eating situation, could be the last drop to make the bucket overflow. In this eating situation, overdemanding is clearly present at the bottom layer of the BOAM basic model, namely through overstimulation of the neurological system. In other situations, the parents also clearly recognised the overdemanding on the second layer of the BOAM basic model because practical changes or rapid social interactions overload her ordering processes. On the third layer of the BOAM basic model, parents recognised that her autonomy is over-demanded in certain tasks that she should be able to do given her age, such as choosing her clothes, dressing herself, or when she depends on Alexander to make contact. With model 5, they came to the conclusion that it was the imbalance in ordering load and ordering capacity that caused her frequent loss of self-regulation. The parents recognised that her ordering load was chronically quite high in the busy family, the school type without fixed structures, and the fast-paced society. They believe she experiences a combination of sensory overload and an excess of information, as well as quick changes and choices. Model 4 was then used again to assess whether other disturbances (neglect, trauma or indirect trauma) also play a role. The parents did not recognise trauma but recognised a specific form of neglect, namely spoiling, in reaction to the loss of self-regulation of Victoria. They tended to follow her in her own solutions to the problems she experienced, even if these solutions were not healthy, instructive or fair to Alexander (who did have to follow certain parenting rules). They realised that this hinders Victoria in learning to deal with situations in which she ‘does not get what she wants’, and this added dysfunctional experiential information to her frame of reference (namely that it works to have tantrums when you want something). The therapist explained that a sequence of pictograms could provide her basic need for structure and predictability and support Victoria in her ordering processes of practical and social situations.

The parents felt that ‘ordering problems’ was not a hypothesis anymore but an appropriate, explanatory diagnosis of their daughter’s behaviour problems. Also, in this session, they specified it as ‘overdemanding-related ordering problems’, which are related to several environmental factors, such as a school without fixed structures. They add spoiling as being a parenting response that worsens these problems, which arose from the powerlessness they have experienced so far, to guide her with her problems in a better way.

Victoria herself played a minor role in this session because her parents had not announced to her that she could not use a digital device during the session; they had already agreed that she could watch videos, and she had already counted on that. Given her difficulties with sudden changes, the therapist decided not to intervene but repeated her request for the next appointment. The homework assignment that was given to the parents for the next time was to make a list of (specific) family rules. The parents were now motivated to do this, while earlier, they associated rules with punishments, which they did not want to use in their parenting.

#### 2.3.5. BOAM Session 4: Model 6 and 8 (Both Parents, Victoria and the Infant, Week 12)

This time, the mother told Victoria beforehand that she could not watch videos. Victoria’s mood seemed positive, and she was less shy than last time. The mother stated that she and her husband had a long discussion about the necessity of these rules, and for the first time, she felt that they were on the same page. She said that the number of incidents with Victoria had decreased by half since then. When the infant asked for attention, the mother asked the father to sit with her in the waiting room.

The therapist and the mother looked at the homework while Victoria played. They specifically looked at the rules regarding the meals. Even though the subject was something Victoria did not like (eating regular meals), she seemed positively interested in the structuring of the rules. Sometimes, she reacted to the rules that the mother and therapist were establishing, and her ideas were then integrated. When the mother gave examples of difficulties that occur, the therapist checked with Victoria: ‘I think that you could not understand that your mother wants you to eat vegetables because she knows that is needed to keep your body fit for dancing?’ Victoria nodded in concentration. The therapist pointed at the BOAM basic model and checked, ‘I think that it is not predictable for you how many vegetables you must eat per day for that?’ Victoria nodded. The therapist pointed to the BOAM Effect Model and checked, ‘I think that is the reason why your scales tip the wrong way and you fall out of your tree and feel panic?’ Victoria nodded again. ‘I think you would like your mother to tell you how many spoons of vegetables are exactly needed for a fit body?’ She nodded again. Together, the mother and Victoria started a list of vegetables that were not too difficult for Victoria to eat and put how many spoonfuls she should eat at dinner. The therapist suggested that they elaborate on this at home.

On the initiative of the therapist, the father and the infant joined again so the father could attend the explanation of the following models. The father was happy because he wanted to discuss an incident that took place last week. He and Victoria had to buy sneakers for the school sports class. Victoria preferred a different kind of shoes. However, these shoes were not suitable for sports, and the father bought the sneakers. Outside the shop, she had a tantrum. She continued to sit on the roadway of the parking lot and refused to get into the car until the father bought her the preferred shoes. The father concluded that this was not due to overstimulation because, in that case, she would have been motivated to get in the car and go home. The therapist complimented him on this good analysis and picked up Model 6, the BOAM Ordering Model (see Figure 5 an the explanation in Box 6).

The therapist pointed at the steps of the ordering process and explained what functional and dysfunctional steps in the ordering process might look like. The first step was sensory information processing. In this step, too many stimuli may lead to overstimulation instead of accurate information. The therapist asked the family whether the family recognises this, for example, Victoria becoming overstimulated when she is at a party or at the playground. The parents confirmed this and gave an example, while Victoria listened attentively to this and seemed content. The next step in the ordering process was to make a connection between the new information and the frame of reference. If this connection is not possible because the experiential knowledge required for this is lacking in the frame of reference, this may lead to confusion. The parents recognised that this happens with Victoria if the incoming information is not specific enough or not timely enough, which happens easily in a hectic family. They reminded Victoria of a situation where they had specifically told her about a change in schedule and in which she was still confused when the situation occurred. Victoria affirmed in her own language. Lastly, the therapist explained that if a connection is made between the information and the frame of reference, this may lead to a dysfunctional reality perception if the experiential knowledge (that is unconsciously stored therein) is one-sided, chaotic or negative. The shop situation seems to be an example of problems with this last step in the ordering process. The therapist asked whether Victoria was normally allowed to choose the shoes she liked. This was indeed the case. The therapist suggested that Victoria connected the information ‘I am going to buy shoes’ to the stored information in her frame of reference: ‘when buying shoes in a shop, I can pick the shoes I like’, and thus had a perception of reality that missed the nuance that this time, she could only choose sneakers because they were meant for sports class. The parents confirmed that this indeed might have been the case. The therapist pointed out that the behaviour stemming from overstimulation, confusion or a dysfunctional reality perception is never from unwillingness but from inability.

**Figure 5 ijerph-22-00559-f005:**
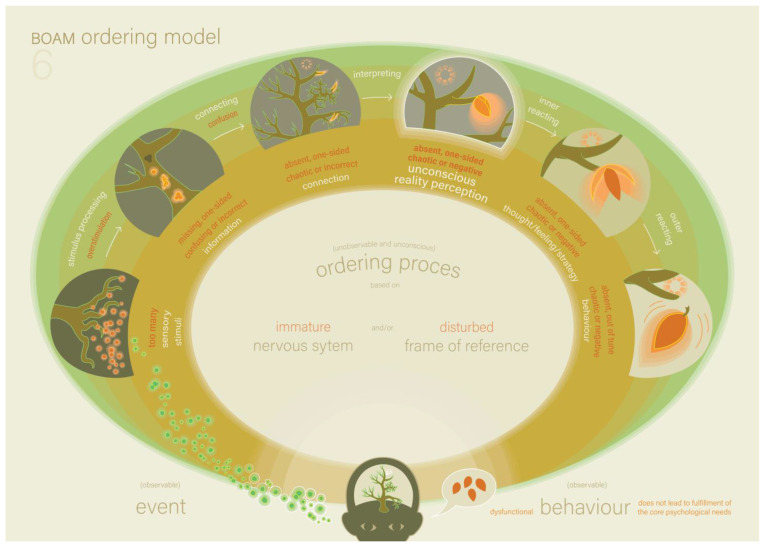
The BOAM Ordering Model (Model 6) [16].

Box 6The BOAM Ordering Model (Figure 5).Model 6 shows the ordering process step-by-step and illustrates both functional and dysfunctional ordering at every step. Over-stimulation can occur instead of sensory information processing. This happens when the neurological nervous system becomes overstimulated due to an excess of stimuli. Confusion can occur instead of reality perception. This happens when the person cannot connect new information because the necessary experiential knowledge to do so is not available in the frame of reference. These problems in the first two steps of the ordering process are quite easy to spot because the person does not arrive at a reality perception, causing a lack of basic safety. Problems in the next steps of the ordering process are less easy to spot because the person does arrive at a reality perception, and thus has a direct experience of basic safety because the core need for ordering is fulfilled. However, if that reality perception does not sufficiently match (other’s) reality, the behaviour strategy based on it will not lead to the desired practical or social goal, and thus indirectly lead to a psychological failure experience. When someone can only link new information to ‘a frame of reference with absent, one-sided, chaotic or negative experiential knowledge and/or negative core beliefs’, this person unconsciously acquires ‘an absent, one-sided, chaotic or negative reality perception’, which will evoke corresponding thoughts and emotions, which then leads to corresponding dysfunctional behaviour.

The therapist then took Model 8, the BOAM Behaviour Model, and showed it to the parents (see Figure 6, and the explanation in Box 7). Victoria was playing with blocks. The therapist explained that this model might be helpful in understanding the part of the shoe-buying situation in which Victoria had a tantrum in the parking lot. The therapist and the parents reconstructed and analysed this situation with the help of Model 8. The primary problem was the fact that there was an ordering problem because of Overdemanding: Victoria’s perception of reality did not match the reality (of going to the shop to buy sneakers and no other type of shoes). She first tried to convince her father that this was not what she expected to happen (primary behaviour). Her father, not realising something had gone wrong in her ordering process, reacted as if Victoria was unwilling and just whining to get her way (secondary problem). He did not respond and bought the required sneakers. While still in the shop, surrounded by other people, Victoria deployed her self-regulation out of fear of unexpected reactions from unknown people and became very quiet (secondary behaviour). This behaviour made her father think Victoria had accepted the situation. However, when they left the shop and arrived at the parking lot, Victoria’s self-regulation ‘run out’, and she fell prey to externalising (tertiary behaviour). She accused, screamed, demanded, threatened, and refused to step into the car, continuing with this until the father conceded and bought her the desired shoes as an extra pair. By applying the model to this specific incident, the parents now understand how the behaviour problems of Victoria build up in general. Doing this in the presence of the child can result in them experiencing exoneration.

Box 7Model 8: The BOAM Behaviour Model (Figure 6).The BOAM Behaviour Model shows how one’s own behaviour can cause more problems, keeping the original disturbance out of sight and unsolved. There are four forms of human behaviour: original, deployed, harmful and malicious behaviour. The latter does not occur in every human being, but the first three do. In the model, these forms of behaviour are represented by three coloured ‘behaviour circles’ that build up around the human figure, from the inside out and in response to disturbances to ‘survive’ these. The model starts with the primary problem, namely overstimulation, neglect, trauma and/or indirect trauma. Primary behaviour: a child responds to the primary disturbance by natural behaviour that transparently reflects this primary problem. The child signals to the parent that a basic need or the need for ordering is under pressure, which prevents it from experiencing basic safety. Secondary problem: the primary behaviour may be misunderstood or unwanted and causes a dysfunctional or negative reaction in the parents, such as misunderstanding, denial, rejection, fear or aggression. These reactions cause attachment ruptures (that every person suffers to some degree), resulting in a decreased trust within the child that the parents can and will fulfil the basic needs or need for ordering. Spoiling is also a problematic reaction of the parents to the expressed needs of their child. This reaction does not cause attachment ruptures but instead creates dependency and unrealistic self-confidence as the secondary problem. Secondary behaviour: the child will try to behave in a way that parents do understand, expect or desire out of fear or false hope. The child deploys (‘misuses’) self-regulation to deny the authentic needs, feelings, trauma or inability, and instead forces deployed behaviour, such as pleasing, lying, parentification, pretending, etc. Tertiary problem: The consequence of secondary behaviour is that it hides the primary problem, and, at some point, self-regulation will run out. Tertiary behaviour: During the secondary behaviour, the initial needs, thoughts and feelings were being transformed into uncontrollable desires, fixations and emotions. This causes uncontrolled behaviour, which is represented by three dragons that depict externalising behaviour, internalising behaviour and ignoring behaviour (e.g., dissociation or addiction). This behaviour logically induces negative reactions (such as misunderstanding, rejection, aggression or even more spoiling) from the environment, thus repeating the secondary problem, but this time ‘justified’ because of the extreme behaviour. This is called a self-fulfilling prophecy and leads to a vicious circle while the primary problem has gone out of sight. If mental health care is sought, it may be that the focus will be only on the tertiary (and sometimes secondary) behavioural patterns and symptoms, which will never really be resolved for the long term until the primary problem is recognised, acknowledged and addressed.

At the end of this session, the therapist introduced a topic outside the scope of the BOAM trajectory, namely the possibility of a standard diagnostic assessment at the mental health centre. The therapist told the parents that the symptoms and behaviour of some children who have overdemanding-related ordering problems could also be classified with autism spectrum disorder. The therapist explained that when children experience many ordering problems, like children with autism spectrum disorder, they may also act out and externalise or internalise their stress more often than other children. Finally, the therapist explained that a DSM-5 classification can sometimes open doors with regard to finances or special needs possibilities and can give clarity to involved professionals such as a teacher or coach, who may then be more understanding. Before the BOAM trajectory, the parents had made clear that they did not want a DSM classification for Victoria. Now, the parents said that they wanted to think about it.

In this session, the parents and therapist refined the prior diagnosis of overdemanding-related ordering problems by identifying which steps in the ordering process are often problematic for Victoria and how the ordering problems lead to secondary behaviour (trying to behave in a difficult situation by becoming quiet and withdrawing in herself), and that subsequently leads to the tertiary behaviour (shouting, threatening, hurting, etc.).

#### 2.3.6. BOAM Session 5: Models Applied to Parents (Parents and Infant, Week 14)

The mother decided that this time, she did not want to bring Victoria to the session, so the parents and therapist could talk about the collaboration between the parents. The mother said she is frustrated with the father’s parenting. The father was somewhat shocked by the mother’s experience but listened attentively to what she shared. The therapist invited the mother to give a specific example of a situation she feels frustrated about. One example she gave was when she asked the father to take the children upstairs in the evening, which he does, but then goes to play with them instead of performing the evening routine. Then, when the mother comes upstairs, the children are hyperactive and difficult to manage.

As always, the models are on the table, and the mother takes the initiative to understand the father’s behaviour with the help of the models. She suggested that the father may also be suffering from overdemanding-related ordering problems, and the father recognised this. The therapist suggested that the father may need to hear more specific information about what to do (and not just what not to do). Together, they started a structure of the steps that he could take when bringing the children to bed, and the therapist suggested that they elaborate on this at home. The father seemed relieved that his own difficulties were becoming clearer. From this moment, he was more actively involved in the sessions and was asking more questions than before. For the mother’s self-reflection, the therapist suggested model 8. As her primary problem, she recognised that she feels overloaded by family life in combination with work and missing the support of the father in the family. Instead of addressing this problem with him, she tried not to complain and just went ahead with all the tasks (secondary behaviour). She already felt that the father’s behaviour came from inability and not unwillingness and did not know how to address this. However, she noticed that she could not keep this up forever and had already started to lose her self-regulation and act annoyed more often (tertiary behaviour).

In this session, both parents gained more self-knowledge and a better understanding of the other and expressed their confidence that they would be able to apply the models in new situations, also independently at home. The diagnosis was refined again, this time regarding the family system; the father recognised overdemanding related ordering problems also with himself, and the mother recognised signs of being overloaded. The assigned homework was to start using pictograms and a weekly planner.

#### 2.3.7. Session with the Intake Psychologist (Parents, Week 15)

In between the BOAM sessions, the parents had a session with the psychologist, who performed the intake with them to discuss the possibility of a standard diagnostic assessment of Victoria. The mother told the psychologist that things are much better with her at home. Victoria was angry less often and behaved much better now that she experienced more structure and predictability. Alexander was no longer responsible for the regulation of his younger sister. The only situations the mother still finds difficult are with friends because these are more difficult to predict. The father also noticed improvements but still experienced difficult incidents with her and found it difficult to deal with her anger. The psychologist suggested that it may be a good idea to make a step-by-step plan for the father. She stated she would discuss this with the BOAM therapist.

The parents discussed with the psychologist that they changed their minds and wanted a standard diagnostic assessment. They believe that if a specific disorder is classified, this can give more clarity, especially to people around them. They expect that people would understand Victoria’s behaviour better and offer her more structure and predictability. The psychologist confirmed that they would perform the assessments in the alternative week from which the BOAM sessions take place.

#### 2.3.8. BOAM Session 6: Model 10 (Parents and Victoria, Week 18)

For the first time, the mother decided to leave the infant with the babysitter. The intake psychologist announced the plan to create a step-by-step plan for the father to deal with his daughter’s anger. This was a natural occasion to introduce Model 10 (the BOAM Recovery Model), which contains the ‘recovery formula’ (for Model 10, see Potharst et al. [17], and the explanation in Box 8).

Box 8Model 10: The BOAM Recovery Model (not included in this article).The BOAM Recovery Model shows the ‘recovery formula’ that transforms diagnostic insights into a concrete approach for parents in difficult moments, in which children show tertiary ‘dragon-behaviour’ (externalising, internalising or ignoring) or secondary behaviour (pleasing, avoiding, lying, etc.). With this, they can help solve their child or adolescent’s psychological problems themselves, whether or not therapy is needed for their child in addition. Before showing the three steps parents can take to de-escalate and solve the problem, it shows the attitude with which parents should approach their child. This is a ‘giving focus’ rather than falling back on their own secondary (pleasing, spoiling) or tertiary behaviour (externalising, internalising or ignoring) in response to their child’s behaviour. Step 1, Acknowledge: The parents give the child acknowledgement, not only of their feelings but particularly of the primary and secondary problems on the BOAM Behaviour Model because these were created by the environment. For example, the primary problem may be overdemanding (“I understand that this situation was unpredictable for you”) or trauma (“your father and I have had bad fights in the past, while you were present, and these moments and feelings come back to you when we don’t agree with each other”). Step 2 Release: The parents allow the child to discharge its stress and emotions in a harmless way (and do not try to have a conversation with the child meanwhile or solve the situation for as long as the child is not ready for that step yet). Step 3, Fulfil: The parents say sorry for their dysfunctional response to their child’s behaviour that caused the secondary problem (also in the case of spoiling), and this time, they try to fulfil the actual basic need or need for order that were pressured by the primary problem. In case of overdemanding, this fulfilment may involve protection from too many stimuli, changes or performance pressure and/or providing more structure and predictability. In the case of trauma, fulfilment may involve protection from new risks of traumatic experiences and providing positive attention, reassurance and a restored connectedness.

The mother recognised the steps of the recovery formula; she stated that she had already intuitively taken the steps and might use the steps even more effectively now she saw them on the model. The father stated that he does not think he will be capable of applying the model in difficult moments with tantrums. Both parents agree that preventing difficult moments rather than dealing with them should still be the priority because it proved to make a big difference for Victoria and the family, and there still seems to be room for improvement. During this session, Victoria played by herself, in contrast to the previous sessions.

The family showed the board for the pictograms, and the mother said that Victoria wanted to decide which pictograms were put on the board and which were not. It seemed that Victoria associated the pictograms more with ‘having to do what mummy wants me to’ than with supporting her ordering process. This is unfavourable but seems inevitable because not only does the family need to learn to make more use of structuring, but Victoria also needs to learn that the parents decide on the structure and set limits. The therapist’s reaction proceeds according to the recovery formula: First, she spoke loudly to give the playing Victoria acknowledgement (step 1) for the fact that her parents are becoming increasingly strict and supposed that the structure may also have actually confused her at times. Then, the therapist remained silent to give Victoria room to release her emotions (step 2), which she did by silent nodding. Then, the therapist tried to give fulfilment (step 3) to Victoria’s need for predictability when using the pictograms by explaining to the parents that before making a new structure, they should remind Victoria that the first draft of a structure will never be exactly ‘right’ and definitive, but will always have to be adjusted a few times before it ‘fits’. Victoria listened attentively, seemed to understand this and looked satisfied.

Both parents introduced the subject of the evening routine, which could take hours for Victoria and Alexander. They play a lot in between the different steps of the routine. The mother stated that she thinks that it is hard for the father to be consistent with the children. The therapist suggested removing the toys from the bathroom. The mother said that playing is important for children. The therapist agreed with this in general but gave some nuance by saying that playing is not suitable during every moment of the day, for example, when leaving the house, and that the evening routine will be done much faster without playing. The mother was still in doubt regarding this issue. The homework was to make the evening routine visible for every family member by using pictograms.

#### 2.3.9. BOAM Session 7: Making Routines (Parents and Victoria, Week 20)

As the parents talked more freely about their parenting, the father introduced something that he experienced as difficult in the collaboration with the mother. The mother has her own habits with the children and wants to be flexible with them without considering the time that something takes. The father said that this makes it difficult for him to plan things. What also plays a role in the difficulty with planning is that the mother needs to work when she is with the children. The therapist acknowledges both parents’ situations and perspectives, and they talk about the role of structure and demands of society, school and work.

The parents also revisited the subject, stating that not only Victoria but also the father needs very clear and stepwise instructions. The parents made an evening structure for the actions that the children take in the evening routine, and as a homework assignment for the next appointment, they are asked to make a step-to-step structure for the father’s role in the evening routine with the children. Victoria did not need to come to the next session; the BOAM therapist would have an evaluation with the parents.

#### 2.3.10. BOAM Evaluation (Parents, Week 22)

This was the last session of the BOAM trajectory, in which the therapist and parents evaluated the trajectory with the help of the BOAM models. Both parents stated that because of their understanding of Victoria’s ordering problems and her heightened need for structure and predictability, they have made their family life much more structured and predictable. Victoria seems to experience much more basic safety, and her autonomy (especially her self-regulation) has improved. Tertiary behaviour in the form of aggression and oppositional behaviour has decreased. The mother uses the BOAM recovery formula effectively, but the father said he has problems applying it. For the mother, it seems easier to recognise what went wrong in Victoria’s ordering process when Victoria gets angry, and therefore she can give her the proper acknowledgement that she needs to calm down. Because the father also experiences overdemanding-related ordering problems, it is more difficult for him to empathise with his daughter to understand her limited reality perception. This also makes it difficult for him to predict which situations will challenge her ordering processes and possibly lead to tertiary behaviour, so these moments surprise and overwhelm him. He understands that he should not relapse into his secondary behaviour (pleasing, giving in to Victoria’s demands), but he is not able to apply the recovery formula (acknowledging what Victoria experienced as unpredictable, giving space to her negative feelings, and setting a new structure). For now, the parents agreed that the mother would deal with difficult situations. The follow-up appointment with the BOAM therapist would be in four weeks. During that appointment, the parents and therapist would talk about whether more treatment is needed.

At the end of this BOAM trajectory, Victoria’s full explanatory diagnosis is that “*nurture* (ordering load) does not match *nature* (ordering capacity) well enough because of the overdemanding of her neurological system and ordering processes.” This high ordering load is related to several environmental factors: a school without fixed structures, the busy family life, and the fast-paced society she lives in. In moments that the ordering load exceeds her ordering capacity, her autonomy decreases (in particular self-regulation), which increases the number of psychological failure experiences. This is not only a short-term problem but since the frame of reference is built from experiences, it is also unfavourable in the long term because it hinders the development of a functional frame of reference. In an attempt to avoid his daughter’s failure experiences, the father tended to spoil his daughter. This deprived her of essential learning experiences and added dysfunctional experiential information to her frame of reference (namely, that it works to have tantrums when you want something). The father also recognised overdemanding-related ordering problems in himself, which made it more difficult for him to empathise with Victoria’s limited reality perception and her psychological need for structure and predictability. Victoria could hide her ordering problems to some extent with secondary behaviour but oftentimes cannot keep this up and then lapses into non-compliant and aggressive tertiary behaviour. Adding more structure and predictability to her life helps solve her primary problem by improving her ordering processes, thus increasing functional reality perceptions, basic safety and self-regulation, which automatically leads to decreasing her behavioural problems.

#### 2.3.11. Session with the Intake Psychologist (Parents, Week 23)

The intake psychologist conducted the psychological assessment with the parents, during which they also talked about how things were going at home and whether the structures that were made were effective. The mother said that things were already going much better: Victoria was only getting angry two or three times a week instead of two or three times a day. However, Saturday mornings were still difficult because there was no school routine, and the family needed to hurry for Victoria and Alexander’s dance and sports lessons. If the father or Alexander even looked at Victoria or talked to her, she became angry. She gets especially angry if her father says something different than what her mother said or would normally say. The psychologist suggested that the parents hang a large pictogram in the room, which makes clear to all family members that, at this specific (and difficult) moment, only the mother deals with Victoria.

#### 2.3.12. Follow-Up Session with the BOAM Therapist (Parents, Week 26)

The parents shared that they had seen a further improvement in Victoria’s behaviour in the past few weeks. Her self-regulation was still growing, and she did not usually get angry anymore (because her need for order was not fulfilled), even in situations that she experienced as difficult. Her dancing teacher also noted that Victoria was easier to teach now. The mother said that she is capable of dealing with Victoria’s ordering problems and that she has no need for further treatment. She says that the father still found it difficult to implement the new parenting routines sometimes, and the father agreed with this. He would like to have more support with this, which he would ask for at the intake psychologist.

#### 2.3.13. Evaluation with the Intake Psychologist and BOAM Therapist (Parents, Week 30)

The parents were very content with their progress. They now understood the cause of Victoria’s problems well and knew how to solve them. The parents were much more on the same page than before. The mother said that she did not feel happy at all as a mother at the beginning of the trajectory, and she now feels very happy (she went from a 2 to a 9 on a scale from 1 to 10). Because the father still found it difficult to deal with Victoria’s anger, the intake psychologist suggested that parental guidance at home would be a possibility. The parents did not want that because they feared that their parenting philosophy would be questioned. They preferred to continue applying the insights and tips they gained from the BOAM trajectory and see if that was enough. The psychologist would stay in touch with them to see if an additional request for help may arise.

The psychologist and therapist had some worries about the sociocratic school and whether their educational system matches Victoria’s need for structure and predictability. They wondered whether Victoria would be able to show initiative in learning, which is a prerequisite to be offered lessons in this type of education. The parents listened to their worries and said they would speak more about this together.

#### 2.3.14. Period of ‘Finger on the Pulse’ by the Intake Psychologist

The standard diagnostic assessment was finished, and a multidisciplinary meeting was held in which autism spectrum disorder was classified. Because the father was too busy, the psychologist shared this with the mother in week 36. The mother was happy with the diagnosis because it would help them communicate better with the outside world about Victoria’s problems and needs. The mother stated that things were still going well. She had even been able to go away for a weekend with Alexander while Victoria stayed with her father. Victoria was now capable of communicating her own needs. For example, she may ask for 10 min of screen time with a timer when she feels overstimulated. Nine out of 10 difficult situations go well. The mother stated that, for now, the father no longer had a request for more support.

In week 44, the psychologist asked the mother whether the parents were able to read the diagnostic report, whether they had any remarks, and how things were going. The mother replied that she had read the report and agreed with it, and things were still going very well. In week 53, the psychologist saw the parents for the last time. They told her that everything was still going well and that they had no request for further help.

### 2.4. Measures

*Child psychopathology* was measured using the Dutch version [42] of the Child Behaviour Checklist (CBCL) for parents of children aged 6 to 18 years old [43]. Both parents rated the 113 items on a 3-point Likert scale, ranging from 0 (not true) to 2 (very true or often true). Higher scores indicate higher levels of child psychopathology. Good psychometric properties have been shown for the Dutch version of the CBCL [42]. An example of an item is ‘Gets in many fights’. The total score and two broadband syndrome scales, internalising and externalising psychopathology, were used for the current study and calculated by adding the relevant items. Scores on these scales range between 0 and 226, 0 and 64, and 0 and 70, and the cut-off point for scores in the clinical range are >49, >14 and >15, respectively. The mother completed the CBCL at waitlist assessment (week 1: right before the intake, 3 weeks before the start of the BOAM trajectory), at the pretest (week 4: right before the start of the BOAM trajectory), at post-test (week 30: at the end of the BOAM trajectory, after the evaluation with the intake psychologist and the BOAM therapist), at 3-month follow-up (week 44: 3 months after the end of the BOAM trajectory), and 5-month follow-up (week 54: 5 months after the end of the BOAM trajectory, at the moment the dossier in the mental health centre closed the dossier). The father only completed the CBCL at baseline and 5-month follow-up assessment. For the calculation of the reliable change index (RCI), a measure for clinically significant change, test–retest reliabilities and standard deviations of the mean scores of a reference group of girls from the Dutch manual were used [42]. Test–retest reliabilities were ICC = 0.89, 0.83, and 0.87, respectively. The standard deviations were 15.4, 5.4, and 5.0, respectively. On the basis of these values, a difference greater than 11.88, 5.18, and 4.19, respectively, was interpreted as clinically significant (*p* < 0.05).

*Executive functioning* was assessed with the parent-report version of the Dutch version [44] of the Behaviour Rating Inventory of Executive Function (BRIEF) [45] for children aged 5 to 18 years old. The mother rated the 75 items on a 3-point Likert scale, ranging from 1 (behaviour is never observed) to 3 (behaviour is often observed) at the pretest, post-test and 3-month follow-up. Higher scores indicate higher levels of problems in executive functioning. Psychometric properties of the Dutch version of the BRIEF were shown to be satisfactory [44]. An example of an item is: ‘Forgets what he/she was doing’. In the current study, the total scale and the two broadband scales, metacognition and behaviour regulation, were used and were calculated by adding up the relevant items. Scores on these scales range between 72 and 216, 44 and 132, and 28 and 84, and the cut-off points for scores in the clinical range are >160, >100 and >65, respectively. For the calculation of the RCI, the test–retest reliabilities and standard deviations of a reference group of 5- to 8-year-old children from a normative study of the Dutch version of the BRIEF were used [44]. Test–retest reliabilities were ICC = 0.86, 0.84, and 0.95, respectively. The standard deviations were 22.9, 15.6, and 9.9, respectively. On the basis of these values, a difference greater than 19.93, 14.52, and 8.12, respectively, was interpreted as clinically significant (*p* < 0.05).

*Parenting stress* was measured with the Dutch Parenting Stress Index (PSI) short form (SF) [46], which is based on the American Parenting Stress Index [47]. Both parents rated 25 items on a 6-point Likert scale, ranging from 1 (totally disagree) to 6 (totally agree). Higher scores indicate higher levels of parenting stress. The Dutch PSI possesses good reliability and satisfactory validity [46]. An example of an item is ‘Parenting this child is more difficult than I thought it would be’. Scores of the PSI-SF are calculated by adding all items, and scores range between 25 and 150. The cut-off point for very high scores is >89 for mothers and >79 for fathers. Mother completed the PSI-SF at all five measurements and father at baseline and 5-month follow-up. As the manual did not contain test–retest reliability and standard deviations of the mean scores [46], the internal consistency and standard deviations of the mean scores of mothers and fathers of a control group of a large study that included the PSI-SF were used for the calculation of the RCI [48]. The internal consistency of the PSI short form was Crohnbach’s *α* = 0.94 for mothers and 0.95 for fathers, and the standard deviations were 19.48 and 17.74, respectively). On the basis of these values, a difference greater than 11.10 and 9.23 for the mother and father, respectively, was interpreted as clinically significant (*p* < 0.05).

*Partner relationship* was measured by the subscale Partner relation of the Family Functioning Questionnaire (FFQ, in Dutch: Vragenlijst Gezinsfunctioneren voor Ouders) [49]. The FFQ aims to measure different aspects of family functioning. The subscale Partner relationship consists of 5 items that are rated on a 4-point Likert scale, ranging from 1 (does not apply) to 4 (applies completely). Higher scores indicate higher levels of satisfaction with the partner and the level of appreciation and support from the partner. The mother rated the FFQ at pretest, post-test, and 3-month follow-up. An example of an item is ‘I feel supported by my partner in taking care of the children’. Scores are calculated by adding all items, and scores range between 5 and 20. The cut-off point for scores in the clinical range is >12. The psychometric properties of the FFQ are good [49]. Because the manual of the FFQ does not report a test–retest reliability, the internal consistency was used (Crohnbach’s *α* = 0.89), and the standard deviation of the mean score of the reference group of parents of 4- to 11-year-old children (2.6) [37]. On the basis of these values, a difference greater than 2.01 was interpreted as clinically significant (*p* < 0.05).

*Acceptability* of the BOAM trajectory was measured by a 9-item evaluation questionnaire that was completed by the mother at the post-test. The following questions were included in this questionnaire: How do you feel about the improvement that was made? How suitable was the BOAM trajectory for your request for help? To what extent would you recommend BOAM to other families? How do you feel about what you have learned as a parent? How suitable was the BOAM method for your referral? How content were you with the therapist? To what extent they could use what was learned in daily life? To what extent did your self-knowledge increase? To what extent did your confidence in the future increase?

### 2.5. Outcomes

Table 1 shows all questionnaire scores at all available measurements. The mother-rated baseline CBCL total score and externalising subscale were in the clinical range, and the internalising subscale was in the subclinical range. The father-rated CBCL externalising subscale was a subclinical score, and the total score and internalising scale were in the normal range. The mother-rated BRIEF scale behaviour regulation was in the clinical range, the total score was in the subclinical range, and the scale metacognition was in the normal range. Both parents’ scores on the PSI were categorised as very high. The mother-rated pretest FFQ score was in the normal range.

The scores at the different time points were compared using the RCI. The RCI is a statistical value showing the degree to which change during an intervention is greater than might have occurred just due to measurement error alone, on the basis of which can be determined if clinically significant change has occurred [50]. Baseline scores were compared with the pretest scores to check whether there were clinically significant changes in the period between the intake and the start of the BOAM trajectory. This was not the case. Post-test and 3- and 5-month follow-up scores were compared with baseline scores (or pretest scores if the baseline was unavailable). The mother reported a decrease in her daughter’s total psychopathology and externalising psychology at post-test and 3- and 5-month follow-up as compared to the pretest. She reported a clinically significant improvement in internalising psychopathology at 3- and 5-month follow-up but not at post-test. She did not see a clinically significant decrease in total executive functioning problems or in metacognition, but she did report an improvement in behavioural regulation at post-test and 3-month follow-up (a 5-month follow-up was not available for this questionnaire). The mother experienced a decrease in parenting stress at post-test and 3- and 5-month follow-up. The mother did not report an improvement in the partner relationship.

The father only completed the CBCL and PSI at baseline and 5-month follow-up. He reported a clinically significant improvement in total psychopathology and externalising psychopathology at the 5-month follow-up but no improvement in internalising psychopathology. He experienced a decrease in parental stress between baseline and 5-month follow-up.

At the post-test, the mother completed an evaluation form. The first four questions were rated on a scale from 1 to 7. Mother rated all four questions a 7. The last five questions were rated on a scale from 1 to 10. Mother rated all five questions a 10.

## 3. Discussion

This case illustrates the way in which BOAM can be used in families with children with problems in self-regulation. It shows how the BOAM models can help the parents gain an understanding of the (multidimensional) causes of their child’s problems. It shows how, even with young children, some shared understanding and a way to talk about problems emerges. It also illustrates how self-reflection in parents may be stimulated by the models and how parents may translate gained understanding to changes in their everyday lives. In this particular case, the externalising psychopathology, behaviour regulation problems, and parenting stress clinically significantly decreased, and after the BOAM trajectory, there was no need for further treatment anymore.

The influence of a therapist is very limited compared to the influence parents can exert on the psychological problems of a child or young person. After all, they are the attachment figures, role models and daily educators. That is why the BOAM method standardises family diagnostics and family treatment for all children and adolescents with (all sorts of) psychological problems. The method focuses on strengthening the parents’ parenting skills in relation to the specific mental health problems, as well as the family dynamics between them. As a result of a BOAM trajectory, a family diagnosis also emerges. This leads to a tailor-made parenting approach with which the parents can further address the psychological problems of their child or adolescent after treatment. Should follow-up assistance still be needed after BOAM self-diagnosis, this often consists of parental guidance in implementing this parenting approach and/or follow-up family counselling. Only if it is unavoidable and serves a clear purpose will additional individual treatment be provided for the child or adolescent. The parents of this case report also expressed the wish that their daughter would not receive individual treatment. They said that it was especially important for them to understand their daughter’s problems better. The BOAM method seemed to be suitable for this family. After gaining an understanding of their daughter’s difficulties, some choices they made as parents shifted, such as changes regarding her schooling.

During the process, parents not only understood their daughter better, but the father, in particular, also gained a better understanding of himself. Also, their perception of a standard psychological assessment changed. When the family admitted the child for mental health care, they were afraid of a stigmatising effect when her behaviour problems were classified after a regular psychodiagnostic assessment. Later on in the process, they saw the benefits of assessment, which resulted in a DSM-5 classification for their daughter after finishing the BOAM trajectory, namely autism spectrum disorder. The father also recognised the BOAM diagnosis of overdemanding-related ordering problems in himself and said he recognised some autistic traits in himself. This is not surprising, given the heritability of autism [51]. The BOAM models helped the father give words to his difficulties, which was important both for himself and for the mother, as well as the way they worked together as parents. The guiding principle of equal collaboration between the parents and the therapist probably also played a role in this case and possibly had a positive effect on the parents, who felt in control of the process, which created a safe space for self-reflection, gaining new perspectives and making new choices.

Another guiding principle was to let the treatment start as soon as possible by integrating this with the diagnostic process and psychoeducation. The parents of the girl already started making changes at home as a result of session 2, especially an increase in structure and predictability, which had an immediate positive effect, namely that the girl listened to their parents and followed their instructions better. In session 4, which was still part of the diagnostic process, the parents told the therapist that the number of incidents had already decreased by half. The last two sessions were focused on treatment recommendations only. The trajectory and the aftercare (existence of a few evaluation and follow-up contacts) were enough for the family; they did not need additional treatment. Eventually, a regular psychodiagnostic assessment was conducted, but the BOAM trajectory had already started, and the high pressure on the family had then already been partly relieved. Meta-analyses also show that psychoeducation can be an effective tool in reducing children’s problematic behaviour in different populations of children, such as children with anxiety or attention deficit [52,53].

The last guiding principle in the development of the BOAM method that should be mentioned is that BOAM should be suitable for families with different cultural and ethnic backgrounds. The family in the current case report was from an Eastern European country, and their mother language differed from the mother language of the therapist. However, they did speak Dutch very well, so there was no language barrier. The mother said that the models fitted very well with what her intuition had already told her but gave more clarity, structure and language to this. The parents had a strong parenting philosophy that was very important to them, and they were afraid that the parenting advice they would receive in mental health care would not be compatible with their philosophy. However, with the understanding they gained from the sessions and some directions from the therapist on how they could translate this understanding into practical solutions, they were able to integrate this within their own parenting philosophy.

A family’s background and culture can influence the degree to which parents can accept a child’s diagnosis. For example, a study on Non-Latina and Latina mothers with a child with autism spectrum disorder showed that Latina mothers struggled with acceptance of their child’s diagnosis and were less able to apply their ASD knowledge to better understand their child needs [54], which seemed to be related to stigma against ASD in the Latino community. However, the fear of stigmatisation is not unique to Latino parents. A systematic review of barriers that parents experience in seeking mental health care for their children showed that the fear of stigma was reported as a barrier in studies from many different countries and cultures [29]. The BOAM diagnostic method may be an alternative for parents who do not feel resistant to general psychological assessment and classification of a disorder, like the family in the current case report at the start of the trajectory.

The daughter of this family was fairly young (6 years old), and she tended to keep some distance from the therapist. Therefore, the therapist did not work directly with the daughter a lot, but she listened while playing and could show whether she agreed with something or not in her own way. The girl regularly listened to explanations of the models and how they were applied to concrete incidents in her own life. Several times, the therapist explained to the parents, and thus indirectly to her, that her behaviour was due to incapability instead of unwillingness and gave useful language to describe this. This helped both the parents and the girl herself to become more aware of situations that were overstimulating or overwhelming for her and the possibility of expressing the offer or need for help. When the therapist and parents worked on new structures to use at home, she also gave her opinion and ideas a few times that were incorporated into the structures, which increased the acceptance of the structures in the home situation.

The mother, in this case study, reported a significant increase in behaviour regulation (but not in metacognition) as measured with the BRIEF, and both parents reported a decrease in externalising behaviour problems (and according to the mother internalising problems to a lesser extent) as measured with the CBCL. Interestingly, the BOAM trajectory did not focus directly on improving the girl’s behaviour regulation or anger management. According to the BOAM theory, in case of overdemanding-related ordering problems, adding more structure and predictability to a child’s life and protecting the child against too much overstimulation and overdemanding will decrease the ordering problems, and thereby increase autonomy (in the case of this girl especially self-regulation) and decrease secondary and tertiary behaviour (in the case of this girl especially externalising psychopathology). The current case study gives initial support for this theory by showing that after focusing on the above-mentioned factors, executive functioning improved, and externalising behaviour decreased. Although prior research has not looked at ‘ordering problems’, evidence is growing for the importance of sensory processing difficulties in developing mental health problems [55]. Also, executive functioning problems have been shown to be associated with psychopathology in youth [56,57].

Parental stress decreased clinically significantly in both parents. Prior research showed a transactional relationship between parenting stress and child behaviour problems, which suggests that interventions directed at one of the two constructs will eventually show improvement on the other construct as well [58]. The decrease in parenting stress in the family of the current case report was probably not only due to a decrease in regulation problems of the daughter but may also be due to a higher understanding in the parents of the problems of their daughter. With the help of the models, the parents made their daughter’s anger and aggression less unpredictable for them. Also, they gained insight into how to influence their daughter’s problems, which made them more manageable and decreased their feeling of being powerless. Indeed, a meta-analysis showed that parental self-efficacy is negatively related to parenting stress [59]. The mother did not report an improvement in the partner relationship between the parents. There is a discrepancy between the questionnaire and what was reported verbally. On the questionnaire, the mother scored very high in partner relationship at the pretest, and this did not change clinically significantly at the post-test or 3-month follow-up. On the other hand, she did verbally report problems in the collaboration between parents during the sessions, and later on in the trajectory, she reported at least some improvement in their collaboration.

This study has several strengths and limitations. A strength is that the case description was detailed, and many of the BOAM models were provided, making it possible for mental health professionals to use aspects of the BOAM method. Another strength is that, a waitlist assessment was available for some measures. The fact that there was no clinically significant improvement between the waitlist assessment and the pretest suggests that the improvement that was found between the pretest and post-test may be attributed to the BOAM trajectory and not to non-specific factors such as the passing of time. A limitation of this study is the fact that because it is a case report, it is not possible to draw conclusions on the general effectiveness of a BOAM trajectory or on the generalisability of the results. Furthermore, the scientific value of this case report is limited because of the design that was used. A methodological improvement would have been to use a single-case experimental design with multiple measurements per study period [60]. Another limitation was the fact that questionnaire data were not available for all measurement points for both parents, and no other informants were included.

There is another limitation that should be mentioned, which is not about the study methodology but the BOAM method. Although theoretically, there are many advantages to this method, especially the collaborative character and the fact that a family or client comes to a ‘self-diagnosis’, the question should be asked how reliable and valid such a diagnosis is. In their exploration of collaborative diagnosis, Hackmann et al. [30] propose that diagnosis can be seen as two overarching elements, namely the diagnostic process and the resulting diagnostic label, and they state that for the patient, there seem to be many advantages to the process of collaborative diagnosis, but that there may be a field of tension with the clinician feeling the need to come to a diagnosis that is most valid. The reliability of diagnoses that are based on collaboration between the family and the therapist may be lower than, for example, observation methods. A meta-analysis on the classification of autism by the Autism Diagnostic Interview—Revised (ADI-R) and the Autism Diagnostic Observation Schedule-2 (ADOS-2) showed that the ADOS-2 had a higher sensitivity and specificity than the ADI-R [61]. On the other hand, a study on self-diagnosis concluded that self-reported diagnoses correspond well with symptom severity on a continuum and can be trusted as clinical indicators, especially with common internalising disorders [62]. Studies on the reliability and validity of collaborative diagnosis and self-diagnoses generally see the diagnosis as a DSM-5 classification. As the BOAM method does not work with symptom-descriptive DSM-5 classifications, it may be more difficult to study the reliability and validity of a BOAM diagnosis. According to BOAM theory, the validity of a BOAM diagnosis is ensured in four ways: (1) The family members answer diagnostic questions on the basis of psycho-education constructed from scientific knowledge elements from (developmental) psychology, neuro-psychology, pedagogy and sociology. (2) The family members are experts by experience. (3) The diagnostics are performed by several family members who also mirror each other for a multidimensional picture. (4) Both the working hypotheses during the process and the final diagnosis arise under the guidance and direction of the therapist. (5) The diagnosis is not aimed at “establishing a symptom descriptive disorder”, but at “providing a plausible narrative that logically explains the mental health problems and leads to a practical approach and realistic hope”.

The BOAM method offers family diagnostics and family treatment for children and adolescents with mental health problems. The models can also be used in a preventive way, for example, in schools. The first therapist’s manual focuses on youth care and will be published in the Netherlands in December 2025. Both quantitative and qualitative methods could give more clarity on the effectiveness of the BOAM method and the way that parents and children experience a BOAM trajectory. Examples of the questions to be answered are as follows: What is the effectiveness of the BOAM method on child, parenting and family outcomes? For what kind of children and families is BOAM of added value? Does a BOAM trajectory at the start of treatment decrease the number of treatment sessions needed? What are the working mechanisms of BOAM? Can an improved understanding of ordering problems in parents predict an improvement in parenting behaviour and in child behaviour problems? What is the meaning of a BOAM diagnosis for a child and parents as compared to a more general diagnosis?

## 4. Conclusions

The current case report illustrated how the BOAM method can be used in families with children with problems with self-regulation and mental health problems by painting a vivid picture of how family conversations can be structured and targeted using the models. The problems in child self-regulation, child externalising behaviour problems and parental stress decreased after the BOAM method was offered, and the improvements were still clinically significant several months later. This case study justifies the need for further evaluation of the BOAM method.

## Figures and Tables

**Figure 2 ijerph-22-00559-f002:**
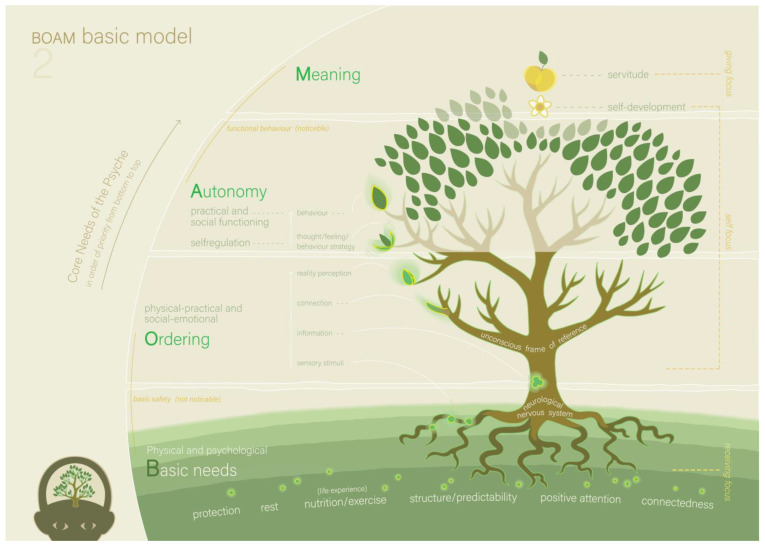
The BOAM Basic Model (Model 2) [16].

**Figure 6 ijerph-22-00559-f006:**
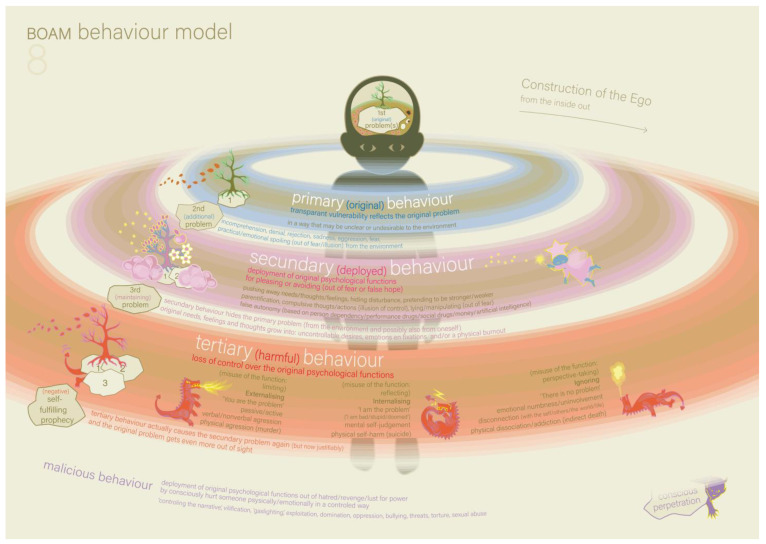
The BOAM Behaviour Model (Model 8) [16].

**Table 1 ijerph-22-00559-t001:** Questionnaire scores at all available time points.

	Mother					Father	
	Baseline	Pretest	Post-Test	3-Month Follow-Up	5-Month Follow-Up	Baseline	5-Month Follow-Up
CBCL total score	78	73	42 *	48 *	51 *	29	8 *
CBCL internalising	12	11	7	5 *	6 *	3	1
CBCL externalising	35	35	17 *	20 *	22 *	13	5 *
BRIEF total score	-	156	137	142	-	-	-
BRIEF metacognition	-	84	79	83	-	-	-
BRIEF behaviour regulation	-	72	58 *	59 *	-	-	-
PSI parenting stress	107	104	79 *	76 *	73 *	105	89 *
FFQ partner relationship	-	19	19	18	-	-	-

- Score unavailable. * Clinically significant improvement, as calculated with the reliable change index, as compared to the baseline (or compared to pretest if the baseline was unavailable).

## Data Availability

The data presented in this study are available on request from the corresponding author. The data are not publicly available due to ethical reasons (personal data).

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
