# Peer review of "A Case Report on How BOAM Offers a Brief Family-Based Treatment by Integrating Psychoeducation and Self-Diagnostics"

_ijerph, 2025, doi:10.3390/ijerph22040559_

Round 1
Reviewer 1 Report
Comments and Suggestions for Authors
Thank you for the opportunity to read and review this manuscript. This manuscript presents a case report of a brief family-based treatment BOAM for a 6-year-old child with emotional and behavioural dysregulation. Overall the manuscript is very well written. The topic is interesting and innovative. The case and the treatment are described clearly with sufficient details. I only have a few minor points that the authors may want to consider.
Introduction:
- From the beginning of the manuscript the treatment has been referred to as BOAM, however only in the last paragraph of the introduction did the authors explain what BOAM stands for. I would suggest presenting the explanation earlier so the readers wouldn’t be left wondering the meaning of it.
- The guiding principles in the BOAM method are described clearly with details. I think it would also be helpful to see a bit more about the BOAM theory, e.g. how this theory was developed and the underlying psychological theories/models.
Measures
- p.21 – ‘Dutch version (CBCL) of the Child Behaviour Checklist’ the brackets should be placed after the title of the scale.
- p.22 Executive Functioniong – It mentioned twice that mother completed the BRIEF at pretest, posttest and 3-month follow-up, which is repetitive and redundant.
- p.22 Partner relationship – Who completed this questionnaire and at what time point? I understand this can be found in Table 1, but I think it would be helpful to also see relevant details in texts.
- p.22 Acceptability - Who completed this questionnaire and at what time point?
- The father completed most measurements less frequently than the mother. Please could the authors provide a brief explanation on why the study was designed in this way?
Author Response
Reviewer 1
Comment 1: Thank you for the opportunity to read and review this manuscript. This manuscript presents a case report of a brief family-based treatment BOAM for a 6-year-old child with emotional and behavioural dysregulation. Overall the manuscript is very well written. The topic is interesting and innovative. The case and the treatment are described clearly with sufficient details. I only have a few minor points that the authors may want to consider.
Response 1: Thank you very much for your kind words and your valuable feedback.
Comment 2 (Introduction): From the beginning of the manuscript the treatment has been referred to as BOAM, however only in the last paragraph of the introduction did the authors explain what BOAM stands for. I would suggest presenting the explanation earlier so the readers wouldn’t be left wondering the meaning of it.
Response 2: Thank you for your suggestion. We have created a new second paragraph of the Introduction, just after the sentence in which BOAM was mentioned for the first time in the Introduction. This new paragraph mainly contains information that was offered in the last paragraph on the introduction in the last version of the manuscript. In this new paragraph on page 2, we start with explaining what BOAM stands for: “BOAM is an acronym for the four core psychological needs of every human being in hierarchical order: Basic Needs, Ordering, Autonomy and Meaning. Ordering is a new term for the unconscious process in which the psyche converts the constant stream of sensory stimuli into a personal perception of reality, in response to which the thoughts, feelings and behaviours arise. Fulfilling these four core needs is seen as the core task of the psyche in BOAM theory, that connects knowledge elements from psychology, neuropsychology, pedagogy and sociology in a relatively simple way. Based on this theory, children, adolescents and their parents receive psychoeducation about psychological processes, during treatment. This is done using models, which are mindmaps with pictures with clear imagery and minimal text. The principles of the development of the BOAM method are explained in more detail next.”
Comment 3 (Introduction): The guiding principles in the BOAM method are described clearly with details. I think it would also be helpful to see a bit more about the BOAM theory, e.g. how this theory was developed and the underlying psychological theories/models.
Response 3: We agree that it is important to give some more information about how the theory was developed and what the underlying theories are. To the paragraph that we made in response to your first comment on page 2, we added the following sentences: “The BOAM-theory and models were developed by author DT, while she was working with families with children with complex problems. In creating the models, she found inspiration in several psychological theories, models and treatments, such as Maslow’s hierarchy of needs model [19], Dawson and Guare’s executive functioning coaching model [20], schematherapy [21], cognitive behavioural therapy [22], and non-violent resistance in families [23]. The principles of the development of the BOAM method are explained in more detail next.”
Comment 4 (Measures): p.21 – ‘Dutch version (CBCL) of the Child Behaviour Checklist’ the brackets should be placed after the title of the scale.
Response 4: Thank you for reading our manuscript so carefully and noticing this mistake. We have now corrected it.
Comment 5 (Measures): p.22 Executive Functioniong – It mentioned twice that mother completed the BRIEF at pretest, posttest and 3-month follow-up, which is repetitive and redundant.
Response 5: Thank you again for your careful reading, we have also corrected this.
Comment 6 (Measures): p.22 Partner relationship – Who completed this questionnaire and at what time point? I understand this can be found in Table 1, but I think it would be helpful to also see relevant details in texts.
Response 6: You are right that we have included this information in the description of the other measures but not in this one, which is inconsistent. We have now included the following sentence to the paragraph describing the FFQ on page 24: “The mother rated the FFQ at pretest, posttest, and 3-month follow-up.”
Comment 7 (Measures): p.22 Acceptability - Who completed this questionnaire and at what time point?
Response 7: Thank you for pointing out that we also did not include this information. We have extended the sentence on the acceptability questionnaire on page 24 as follows: “Acceptability of the BOAM trajectory was measured by a 9-item evaluation questionnaire that was completed by the mother at posttest.”
Comment 8 (Measures): The father completed most measurements less frequently than the mother. Please could the authors provide a brief explanation on why the study was designed in this way?
Response 8: Thank you for pointing out that the reason of the missing data of the father was unclear. We added another paragraph, containing this and some extra information to the section Datailed Case Description on page 5: “This study was approved by the ethical review board of the University of Amsterdam ( 2017-CDE-8422 and FMG-11997-2024). The parents provided written informed consent when they were included in the study and again after reading this case report. Part of the questionnaires that were used in this study were derived from the pretest and posttest of the general effectiveness study of the mental health care centre that the girl was admitted to. Both parents completed these questionnaires before the intake and when the file was closed. In the current case report, these to measurement occasions are called the waitlist assessment and the 5-month follow-up assessment. Furthermore, both parents were asked to complete questionnaires that were part of the BOAM-study. At the time, father was too busy for this, but mother did complete these questionnaires at three time points (in the current study called the pretest, posttest and 3-month follow-up assessments).”
Reviewer 2 Report
Comments and Suggestions for Authors
Please see the attachment.

Author Response
Reviewer 2
Comment 1: The authors have chosen a case of a 6-year-old girl to demonstrate the application of the BOAM method, but they did not provide a detailed explanation of why this case was selected and whether it is representative.
Response 1: Thank you for you’re the time and effort that you spend on reading our manuscript and thank you for your comments, which have helped us improve the manuscript. We have extended the paragraph about the reasons for selecting this case on page 5 as follows: The focus of this case report is on the 6-year-old girl Victoria with emotional and behavioural problems and on her parents. Their case was chosen to illustrate how the BOAM method can be used in collaboration with parents and children, even if they are still young, and to illustrate how the BOAM method integrates family-based diagnostics, psychoeducation, and treatment, thereby making it unnecessary to give children individual treatment. This case was also chosen because it is a representative example of how the focus on development of greater self-insight and empathy in parents, based on universal psychoeducation, can activate their self-solving abilities and strengthen parenting skills. Furthermore, there is a need for interventions for children that are personalized, and that consider the unique perspectives and needs of neurodiverse children [16], what this girl was shown to be later on in the trajectory. Lastly, this case describes a BOAM-trajectory of regular length. Our first pilot study on the BOAM-method focused on families with children who were non-respondent to regular mental health care and often needed a longer trajectory [17]. The family in this case report had received no previous mental health care.”
Comment 2: It is recommended that the authors include a detailed description of the questionnaires in the "Measurement" section, including specific items, scoring criteria, and reliability and validity, to ensure data transparency and reproducibility.
Response 2: Thank you for your comment. We have now made sure that for every questionnaire, we included example questions of every questionnaire in the description (we moved the questions of the evaluation questionnaire from the results section to the measures section), scoring criteria, and psychometric properties on page 23 and 24 (additions are highlighted).
Comment 3: The authors reported changes in questionnaire scores for both the mother and father at multiple time points, but the explanations for these changes are somewhat brief and do not delve into the specific relationship between these changes and the BOAM method.
Response 3: Thank you for your comment. We tried to clarify what was done in the BOAM trajectory and the possible impact on the outcome measures by adding several sentences (marked yellow in the manuscript) to the paragraphs that discussed these outcomes on page 27 and 28: “The mother in this case report reported a significant increase in behaviour regulation (but not on metacognition) as measured with the BRIEF, and both parents reported a decrease in externalizing behaviour problems (and according to the mother internalizing problems to a lesser extent) as measured with the CBCL. Interestingly enough, the BOAM-trajectory did not focus directly on improving the girl’s behaviour regulation or anger management. According to the BOAM theory, in case of overdemanding-related ordering problems, adding more structure and predictability to a child’s life and protect the child against too much overstimulation and overdemanding will decrease the ordering problems, and thereby increase autonomy (in the case of this girl especially self-regulation) and decrease secondary and tertiary behaviour (in the case of this girl especially externalizing psychopathology). The current case study gives initial support for this theory by showing that after focusing above mentioned factors, executive functioning improved and externaliziong behaviour decreased. Although prior research has not looked at ‘ordering problems’, evidence is growing for the importance of sensory processing difficulties in developing mental health problems [56]. Also, executive functioning problems has been shown to be associated with psychopathology in youth [57,58].
Parental stress decreased clinically significant in both parents. Prior research showed a transactional relationship between parenting stress and child behaviour problems, which suggests that interventions directed on one of the two constructs will eventually show improvement on the other construct as well [59]. The decrease in parenting stress in the family of the current case report was probably not only due to a decrease in regulation problems of the daughter, but may also be due to a higher understanding in the parents of the problems of their daughter. With the help of the models, the parents made their daughter’s anger and aggression less unpredictable for them. Also, they gained insight on how to influence their daughters problems, which made them more manageable, and decreased their feeling of powerless. Indeed, a meta-analysis showed that parental self-efficacy is negatively related to parenting stress [60]. Mother did not report an improvement in the partner relationship between parents. There is a discrepancy between the questionnaire and what was reported verbally. On the questionnaire, mother scored already very high in partner relationship at pretest, and this did not change clinically significantly at posttest or 3-month follow-up. On the other hand, she did verbally report problems in the collaboration between parents during the sessions, and later on in the trajectory reported at least some improvement in their collaboration.”
Comment 4: Some references are relatively outdated and do not adequately reflect the latest research developments in the field.
Response 4: You are right that some reference were relatively outdated. We added about 30 new references (especially in new text but also to replace some of the older references). We highlighted the new references in the reference list on page 30 to 33.
Comment 5: There are instances in the article where the language is not sufficiently accurate or fluent, and the structure of some paragraphs is unclear.
Response 5: thank you for your feedback. We have checked the whole manuscript for accuracy of language and a clear structure of paragraphs, and made changes throughout the manuscript, that we marked. Especially in the boxes, we made changes to clarify the explanations we give there.
Reviewer 3 Report
Comments and Suggestions for Authors
Dear Authors,
Thank you for the opportunity to review your article. Your study offers a valuable contribution by exploring an innovative family-based diagnostic approach. Below, we provide constructive feedback to enhance clarity, depth, and impact. We hope these insights assist in refining your work and strengthening its contributions.
Title
A Case Report on How BOAM Offers a Brief Family-Based Treatment by Integrating Psychoeducation and Self-Diagnostics
Abstract
- The abstract effectively covers the topic, objectives, methodology, key findings, and conclusions.
- Question: Does the abstract provide sufficient information about the instruments used? Would it be helpful to include additional details on methods and statistical analyses?
- Lacks more detail on the instruments used and the impact of the findings. Suggestion: Briefly include the assessment scales employed.
Introduction
- Clearly explains key concepts, including the theoretical foundation of BOAM (lines 36-57).
- Question: Does the introduction clearly establish the research gap this study aims to fill? How does this study differ from similar approaches?
- The research question is not explicitly formulated. Recommendation: Reformulate it at the end of the introduction to emphasize the objectives.
- The study's relevance is well justified but could include more comparisons with traditional diagnostic approaches.
Methodology
- It was somewhat difficult to locate the methodology section. The detailed case description is placed in section 2, but it should be labeled "Methods," with "Detailed Case Description" as section 2.1.
- Well-structured section, including research design and sample characterization.
- Question: Are the participant selection criteria well-defined and justified? Is there sufficient information about the context in which data was collected?
- Sample selection criteria could be more detailed.
- Ethical approval for the study is not clearly indicated.
- Question: Was the study approved by an ethics committee? If so, where is this indicated in the text?
Results
- Well presented, with the use of tables (Table 1, line 720).
- Question: Are the results clearly linked to the research questions and hypotheses formulated in the introduction? Is the description of findings sufficiently detailed?
- Some analyses could be better explained, such as the interpretation of the "reliable change index" (lines 726-735).
- Question: Is the statistical analysis detailed enough to allow study replication?
Discussion
- Relates the findings well with other studies but could cite more recent references (lines 812-839).
- Question: How does this study contribute to the research field? Are there clear practical implications for professionals in the field?
- Study limitations are mentioned but could be more detailed.
- Question: Are there additional limitations that could be discussed, such as methodological issues or limitations in generalizing findings?
Conclusion
- Consistent with the results (lines 841-846).
- Question for the authors: Do the conclusions clearly highlight the study’s contributions? Are there recommendations for future research?
- Does not explicitly present suggestions for future research.
References
- Most references are up to date, but some could be replaced with studies from the last five years.
- Question for the authors: What is the proportion of references published in the last five years? Are there more recent studies that could be included?
Study Relevance
- The study fills a gap by presenting an innovative family-based diagnostic approach.
- Question: How can the study’s findings be applied in clinical or educational practice? Are there clear directions for future research or interventions?
General Conclusion
The article is well-structured and relevant but needs adjustments in the formulation of the research question, methodological details, and a more in-depth exploration of study limitations and future implications.
Author Response
Reviewer 3
Comment 1: Thank you for the opportunity to review your article. Your study offers a valuable contribution by exploring an innovative family-based diagnostic approach. Below, we provide constructive feedback to enhance clarity, depth, and impact. We hope these insights assist in refining your work and strengthening its contributions.
Response 1: Thank you very much for the time and effort you put into reading our manuscript and for your comments, that helped us improve the manuscript.
Comment 2: Abstract
- The abstract effectively covers the topic, objectives, methodology, key findings, and conclusions.
- Question:Does the abstract provide sufficient information about the instruments used? Would it be helpful to include additional details on methods and statistical analyses?
- Lacks more detail on the instruments used and the impact of the findings. Suggestion:Briefly include the assessment scales employed.
Response 2: Thank you for your suggestions. We have added the names of the measures that we used in the abstract: “Mother completed questionnaires on child psychopathology (Child Behaviour Checlist), executive functioning (Behaviour Rating Inventory of Executive Function), parenting stress (Parenting Stress Index) and partner relationship (Family Functioning Questionnaire)…” Furthermore, regarding the impact of findings, we have tried to describe what readers may take from this case report by adding a last sentence to the abstract: “This case report painted a vivid picture of how family conversations can be targeted using the models.”
Comment 3: Introduction
- Clearly explains key concepts, including the theoretical foundation of BOAM (lines 36-57).
- Question:Does the introduction clearly establish the research gap this study aims to fill? How does this study differ from similar approaches?
- The research question is not explicitly formulated. Recommendation:Reformulate it at the end of the introduction to emphasize the objectives.
Response 3: Thank you for your suggestions to explicitly include the study objectives. We now added the following sentence to the last paragraph of the introduction on page 4: “The aim of this case report is 1) to illustrate a BOAM trajectory, including the use of the BOAM models in clinical practice in a child mental health setting, and 2) to study the effect of the BOAM trajectory on the participating family.”
- Comment 4: The study's relevance is well justified but could include more comparisons with traditional diagnostic approaches.
Response 4: In the last version of the manuscript, we had made the choice not to make this comparison because we think that BOAM can be used not necessarily instead of, but also in addition to other diagnostic approaches. However, we do understand that it may be a good idea to shortly make such a comparison. We now added the following paragraph to the Introduction on page 2 and 3: “Generally, when a child is admitted to mental health care, a diagnostic process is the first step. This process often exists of interviews with different informants, questionnaires, tests, and observations, to make an inventory, description, ordering and categorization of the problem behavior, which then usually results in a classification of the problem. Often, a classification system such as the fifth version of the Diagnostic and Statistical Manual of Mental Disorders (DSM-5) is used for this [24]. The transdiagnostic approach identified several problems related to classifying problematic behavior, such as that mental problems are not clearly separable from each other. Also, not only biological, but also psychological and social factors play a role in the development and maintenance of mental health problems, which does not become clear with a classification. Furthermore, a classification is often not enough to guide the treatment process, or to offer answers about the cause the problems, while a diagnostic process is also aimed at explaining the problems. As an answer to some of these issues, several alternative diagnostic systems have been developed, such as the Hierarchical Taxonomy Of Psychopathology (HiTOP) [25] and Psychodynamic Diagnostic Manual (PDM-2) [26]. Both are highly specialized and do not seem to be accessible enough to use in close collaboration between the child, parents and therapist.”
Comment 5: Methodology
- It was somewhat difficult to locate the methodology section. The detailed case description is placed in section 2, but it should be labeled "Methods," with "Detailed Case Description" as section 2.1.
Response 5: We can imagine that it is difficult to locate the methodology section. However, we could not follow your suggestion, due to the specific instructions of the journal regarding a case report, which are stated as follows on the website: “The structure of case reports differs from articles and includes an Abstract, Keywords, Introduction, Detailed Case Description, Discussion, and Conclusions.” However, we have added the following sentence to the introduction of the Detailed Case Description section on page 5, where we also added information on the measurement points that were included in this study, to clarify where this information can be found: “More information on the questionnaires that were administered is included in the 2.4 Measures section, that follows the description of the BOAM trajectory.”
Comment 6:
- Well-structured section, including research design and sample characterization.
- Question:Are the participant selection criteria well-defined and justified? Is there sufficient information about the context in which data was collected?
- Sample selection criteria could be more detailed.
- Ethical approval for the study is not clearly indicated.
- Question:Was the study approved by an ethics committee? If so, where is this indicated in the text?
Response 6: Thank you for your suggestions. We have added information on the approval of the ethics committee in the to the introduction of the Detailed Case Description section on page 5: “This study was approved by the ethical review board of the University of Amsterdam (2017-CDE-8422 and FMG-11997-2024). The parents provided written informed consent when they were included in the study and again after reading this case report.” . Also, we have extended the paragraph about the reasons for selecting this case as follows: The focus of this case report is on the 6-year-old girl Victoria with emotional and behavioural problems and on her parents. Their case was chosen to illustrate how the BOAM method can be used in collaboration with parents and children, even if they are still young, and to illustrate how the BOAM method integrates family-based diagnostics, psychoeducation, and treatment, thereby making it unnecessary to give children individual treatment. This case was also chosen because it is a representative example of how the focus on development of greater self-insight and empathy in parents, based on universal psychoeducation, can activate their self-solving abilities and strengthen parenting skills. Furthermore, there is a need for interventions for children that are personalized, and that consider the unique perspectives and needs of neurodiverse children [41], what this girl was shown to be later on in the trajectory. Lastly, this case describes a BOAM-trajectory of regular length. Our first pilot study on the BOAM-method focused on families with children who were non-respondent to regular mental health care and often needed a longer trajectory [17]. The family in this case report had received no previous mental health care.”
Comment 7: Results
- Well presented, with the use of tables (Table 1, line 720).
- Question:Are the results clearly linked to the research questions and hypotheses formulated in the introduction? Is the description of findings sufficiently detailed?
- Some analyses could be better explained, such as the interpretation of the "reliable change index" (lines 726-735).
- Question:Is the statistical analysis detailed enough to allow study replication?
Response 7: Thank you for your helpful suggestion. We have now added the following sentences to the second paragraph of the Results section on page 25: “The scores at the different timepoint were compared using the RCI. The RCI is statistic value showing the degree to which change during an intervention is greater than might have occurred just due to measurement error alone, on the basis of which can be determined if clinically significant change has occurred [50].”
Comment 8: Discussion
- Relates the findings well with other studies but could cite more recent references (lines 812-839).
Response 8: You are right that some reference were relatively outdated. We added about 30 new references (partly with new text and partly replacing older references). We highlighted the new references in the reference list on page 30 to 33.
Comment 9:
- Question:How does this study contribute to the research field? Are there clear practical implications for professionals in the field?
- Study limitations are mentioned but could be more detailed.
- Question:Are there additional limitations that could be discussed, such as methodological issues or limitations in generalizing findings?
Response 9: Thank you for your suggestion. We have now included a paragraph on the strengths and limitations of the study on page 28: “This study has several strengths and limitations. A strength is that the case description was detailed, and many of the BOAM models were provided, making it possible for mental health professionals to use aspects of the BOAM-method. Another strength is that for some measures, we a waitlist assessment was available. The fact that there was no clinically significant improvement between the waitlist assessment and the pretest, suggests that the improvement that was found between pretest and posttest may be attributed to the BOAM trajectory and not to non-specific factors such as the passing of time. A limitation of this study is the fact that because it is a case report, it is not possible to draw conclusions on the general effectiveness of a BOAM trajectory, or on the generalizability of the results. Furthermore, the scientific value of this case report is limited because of the design that was used. A methodological improvement would have been to use a single case experimental design with multiple measurements per study period [61]. Another limitation was the fact that questionnaire data was not available for all measurement points and for both parents, and no other informants were included.”
Comment 10: Conclusion
- Consistent with the results (lines 841-846).
- Question for the authors:Do the conclusions clearly highlight the study’s contributions? Are there recommendations for future research?
- Does not explicitly present suggestions for future research.
Response 10: We also found this suggestion very helpful. We have added a paragraph on future research on page 29: “More research is necessary to demonstrate the effectiveness of the BOAM method. Both quantitative and qualitative methods could give more clarity on the effectiveness of the BOAM-method and the way that parents and children experience a BOAM-trajectory. Questions to be answered are for example: What is the effectiveness of the BOAM method on both child, parenting and family-outcomes? For what kind of children and families is BOAM of added value? Does a BOAM trajectory at the start of treatment decrease the number of treatment sessions needed? What are the working mechanisms of BOAM? What is the meaning of a BOAM-diagnosis for a child and parents as compared to a more general diagnosis?“ Also, we added a little information to the conclusion, namely that this case report painted a vivid picture of how family conversations can be structured and targeted using the models.
Comment 11: References
- Most references are up to date, but some could be replaced with studies from the last five years.
- Question for the authors:What is the proportion of references published in the last five years? Are there more recent studies that could be included?
Response 11: We added about 30 new references (partly with new text and partly replacing older references). We highlighted the new references in the reference list on page 30 to 33.
Comment 12: Study Relevance
- The study fills a gap by presenting an innovative family-based diagnostic approach.
- Question:How can the study’s findings be applied in clinical or educational practice? Are there clear directions for future research or interventions?
Response 12: We have now added the following sentences to the last paragraph of the discussion on page 29: “The BOAM method offers family diagnostics and family treatment for children and adolescents with mental health problems. The models can also be used in a preventive way, for example in schools. The first therapist's manual focuses on youth care and will be published in the Netherlands in December 2025. Both quantitative and qualitative methods could give more clarity on the effectiveness of the BOAM-method and the way that parents and children experience a BOAM-trajectory. Questions to be answered are for example: What is the effectiveness of the BOAM method on both child, parenting and family-outcomes? For what kind of children and families is BOAM of added value? Does a BOAM trajectory at the start of treatment decrease the number of treatment sessions needed? What are the working mechanisms of BOAM? Can an improved understanding of ordering problems in parents predict an improvement in parenting behaviour, and in child behaviour problems? What is the meaning of a BOAM-diagnosis for a child and parents as compared to a more general diagnosis?”
Comment 13: General Conclusion
The article is well-structured and relevant but needs adjustments in the formulation of the research question, methodological details, and a more in-depth exploration of study limitations and future implications.
Response 13: Thank you again for your constructive feedback!
Reviewer 4 Report
Comments and Suggestions for Authors
I would like to sincerely thank you for your work on this case report. Your research is truly fascinating and offers a valuable contribution to the field. The integration of psychoeducation and self-diagnostics in a family-based treatment framework is particularly compelling. That being said, I would like to offer several constructive suggestions that may help improve the clarity and impact of your manuscript:
- The quality of the images included in the manuscript is suboptimal. Would it be possible to insert them in higher resolution to enhance readability and visual clarity?
- The acronym "BOAM" should be explained upon its first occurrence in the text to ensure clarity for the readers.
- A more detailed definition and explanation of "psychoeducation" would be beneficial. This would help ensure that all readers fully understand the concept as intended by the authors.
- (Lines 65-74): The discussion on collaborative diagnosis is relevant, and while I agree with the significance of this approach, further elaboration would be valuable. For example, some tools like the ADI-R have shown limited effectiveness statistically. A more tempered discussion acknowledging potential limitations would strengthen the argument.
- (Line 79): The mention of self-diagnosis lacks citations. Providing references that discuss the validity and implications of self-diagnosis would be helpful, as not all readers may be familiar with this concept.
- The introduction is quite lengthy. Would it be possible to subdivide it into sections with subheadings to improve readability?
- (Pages 93-94): It would be valuable to discuss how parenting norms and values align with or challenge children's rights. Some educational values and norms contradict children's rights, raising ethical concerns. Also, the manuscript briefly touches upon parental resistance to diagnosis. This is an important cultural factor, particularly given significant differences in clinical psychology practices between Northern and Southern Europe. A discussion on cultural relativism in parenting and diagnostic acceptance would enhance the manuscript.
- Formatting Issues: Line 132 & 374: Extra space present (?)
- (Lines 127-128): The term "emotional and behavioral problems" appears multiple times. Could you specify what particular issues you are referring to?
- (Line 156): The child is introduced at this point, but they are referenced multiple times earlier in the manuscript. It may be useful to indicate earlier that further details will be provided in later sections.
- (Lines 466-467): The discussion on autism heredity is somewhat implicit. A more explicit discussion in the main text or discussion section would improve clarity.
- The discussion section is too brief given the breadth of potential topics. As previously mentioned, several aspects could be explored in greater depth, such as comparisons with other tools and methodologies. For instance, Asperger emphasized the importance of free conversation with autistic children in contrast to standardized tools like ADOS or ADI-R. There is also a lack of references in this section. Expanding the discussion with relevant citations would add academic rigor. The conclusion is also too concise. It would be beneficial to include a discussion of the study’s limitations, future research perspectives, and potential improvements in methodology.
- I'm also not sure about the issue of conflicts of interest. Did the researchers create this tool? If so, this point should be presented and discussed.
Once again, I would like to commend you on your work, as this case report makes an important contribution to the field. I look forward to seeing how the manuscript evolves with these refinements.
Author Response
Review 4
Comments and Suggestions for Authors
Comment 1: I would like to sincerely thank you for your work on this case report. Your research is truly fascinating and offers a valuable contribution to the field. The integration of psychoeducation and self-diagnostics in a family-based treatment framework is particularly compelling. That being said, I would like to offer several constructive suggestions that may help improve the clarity and impact of your manuscript:
Response 1: Thank you very much for the time and effort spent on reviewing our manuscript and for your constructive feedback, that has helped us improve our manuscript.
Comment 2: The quality of the images included in the manuscript is suboptimal. Would it be possible to insert them in higher resolution to enhance readability and visual clarity?
Response 2: Thank you for pointing this out to us. We will send all the original images to the editorial office separately from the manuscript, to ensure that they have the images in optimal quality.
Comment 3: The acronym "BOAM" should be explained upon its first occurrence in the text to ensure clarity for the readers.
Response 3: Thank you for your suggestion. We have created a new second paragraph, just after the sentence in which BOAM was mentioned for the first time in the Introduction on page 2. This new paragraph mainly contains information that was offered in the last paragraph on the introduction in the last version of the manuscript. In this paragraph, we start with explaining what BOAM stands for: “BOAM is an acronym for the four core psychological needs of every human being in hierarchical order: Basic Needs, Ordering, Autonomy and Meaning. Ordering is a new term for the unconscious process in which the psyche converts the constant stream of sensory stimuli into a personal perception of reality, in response to which the thoughts, feelings and behaviours arise. Fulfilling these four core needs is seen as the core task of the psyche in BOAM theory, that connects knowledge elements from psychology, neuropsychology, pedagogy and sociology in a relatively simple way. Based on this theory, children, adolescents and their parents receive psychoeducation about psychological processes, during treatment. This is done using models, which are mindmaps with pictures with clear imagery and minimal text. The principles of the development of the BOAM method are explained in more detail next.”
Comment 4: A more detailed definition and explanation of "psychoeducation" would be beneficial. This would help ensure that all readers fully understand the concept as intended by the authors.
Response 4: We have included an extra sentence on psychoeducation, after mentioning this for the first time on page 2. “Psychoeducation means that the child and/or their families are offered information, advice, and coping strategies regarding their mental health and mental health problems, maintain factors and potential interventions [16].”
Comment 5: (Lines 65-74): The discussion on collaborative diagnosis is relevant, and while I agree with the significance of this approach, further elaboration would be valuable. For example, some tools like the ADI-R have shown limited effectiveness statistically. A more tempered discussion acknowledging potential limitations would strengthen the argument.
Response 5: Thank you for your thoughts on this subject. We have added the following sentences to the fourth paragraph of the Introduction on page 3 “In the literature, some information can be found on collaborative diagnosis, in which the client is actively involved and feels valued [30]. It has been shown that clients find both the process of collaborative diagnosis and the resulting diagnosis more meaningful, informative and useful [30]. It has also been suggested that collaboration could mitigate some of the reported negative consequences of diagnosis, such as feeling stigmatised and disempowered [30]. Another approach that is interesting in this regard is therapeutic assessment, which is aimed at providing the client with therapeutic benefits from the assessment process itself and change their narrative about themselves and their environment to a more positive one [31]. However, there is very little evidence for this approach [31].
Also, we added the following paragraph to the Discussion on page 28: “There is another limitation that should be mentioned that is not about the study methodology, but the BOAM method. Although theoretically, there are a lot of advantages to this method, and especially the collaborative character and the fact that a family or client comes to a ‘self-diagnosis’, the question should be asked how reliable and valid such a diagnosis is. In their exploration of collaborative diagnosis, Hackmann et al. [30] propose that diagnosis can be seen as two overarching elements, namely the diagnostic process and the resulting diagnostic label, and they state that for the patient there seem to be a lot of advantages to the process of collaborative diagnosis, but that there may be a field of tension with the clinician feeling the need to come to a diagnosis that is most valid. The reliability of diagnoses that are based on collaboration between the family and the therapist may be lower than for example observation methods. A meta-analysis on the classification of autism by the Autism Diagnostic Interview-Revised (ADI-R) and the Autism Diagnostic Observation Schedule-2 (ADOS-2) showed that the ADOS-2 had a higher sensitivity and specificity than the ADI-R [62]. On the other hand, a study on self-diagnosis concluded that self-reported diagnoses correspond well with symptom severity on a continuum and can be trusted as clinical indicators, especially with common internalizing disorders [63]. Studies on the reliability and validity of collaborative diagnosis and self-diagnoses generally see the diagnosis as a DSM-5 classification. As the BOAM-method does not work with symptom descriptive DSM-5 classifications, it may be more difficult to study the reliability and validity of a BOAM-diagnosis. According to BOAM theory, the validity of a BOAM-diagnosis is ensured in four ways: is ensured in four ways. 1) The family members answer diagnostic questions on the basis of psycho-education constructed from scientific knowledge elements from (developmental) psychology, neuro-psychology, pedagogy and sociology. 2) The family members are experts by experience 3) The diagnostics are done by several family members who also mirror each other for a multidimensional picture 4) Both the working hypotheses during the process and the final diagnosis arise under the guidance and direction of the therapist, 5) the diagnosis is not aimed at “establishing a symptom descriptive disorder”, but at “providing a plausible narrative that logically explains the mental health problems and leads to a practical approach and realistic hope”.”
Comment 6: (Line 79): The mention of self-diagnosis lacks citations. Providing references that discuss the validity and implications of self-diagnosis would be helpful, as not all readers may be familiar with this concept.
Response 6: Thank you for pointing this out to us. We have added the following sentences to the sixth paragraph of the Introduction on page 3: In the literature, self-diagnosis refers to a growing phenomenon that individuals, especially adolescents, are diagnosing themselves with a mental disorder, as a way to understand themselves better [33]. These diagnoses usually take the form of a classification as described in the Diagnostic and Statistical Manual of Mental Disorders (DSM-5). In BOAM-theory, self-diagnosis refers to a descriptive and explanatory diagnosis, that gives clarity on the both the form of expression of the mental problems, but also the causes and interplay with the family and broader environment.
Comment 7: The introduction is quite lengthy. Would it be possible to subdivide it into sections with subheadings to improve readability?
Response 7: Thank you for your suggestion. We have now added four subheadings to the introduction to improve its structure and readability.
Comment 8 (Pages 93-94): It would be valuable to discuss how parenting norms and values align with or challenge children's rights. Some educational values and norms contradict children's rights, raising ethical concerns. Also, the manuscript briefly touches upon parental resistance to diagnosis. This is an important cultural factor, particularly given significant differences in clinical psychology practices between Northern and Southern Europe. A discussion on cultural relativism in parenting and diagnostic acceptance would enhance the manuscript.
Response 8: Yes, you are right that sometimes parental norms and values can contradict children’s rights. We have added two sentences to the subject of parental norms and values about how the therapist deals with this on page 4: “A side note is that some parenting norms and values might contradict children’s rights. With the help of the model depicting children’s basic needs, the therapist can make this a topic of conversation”.
We hope that we integrated your feedback by adding the following paragraph to the discussion on page 27: “A family’s background and culture can influence the degree to which parents can accept a child’s diagnosis. For example, a study on Non-Latina and Latina mothers with a child with autism spectrum disorder showed that Latina mothers struggled with acceptance of their child’s diagnosis and were less able to apply their ASD knowledge to better understand their child needs [54], which seemed to be related to stigma against ASD in the Latino community. However, the fear of stigmatization is not unique for Latino parents. A systematic review on barriers that parents experience in seeking mental health care for their child showed that the fear of stigma was reported as a barrier in studies from many different countries and cultures [55]. The BOAM diagnostic method may be an alternative for parents who do not feel resistant to general psychological assessment and a classification of a disorder, like the family in the current case report at the start of the trajectory. “
Comment 9: Formatting Issues: Line 132 & 374: Extra space present (?)
Response 9: Thank you for your careful reading. We have deleted extra spaces.
Comment 10: (Lines 127-128): The term "emotional and behavioral problems" appears multiple times. Could you specify what particular issues you are referring to?
Response 10: We added the following sentence to the first paragraph of the Detailed Case Description on page 5: “The emotional and behavioural problems that were present were daily recurring anger, temper tantrums, aggression towards family members, and tics.”
Comment 11: (Line 156): The child is introduced at this point, but they are referenced multiple times earlier in the manuscript. It may be useful to indicate earlier that further details will be provided in later sections.
Response 11: The first time the girl is mentioned on page 2, we let the reader know that further details will be provided further on in the text: “The current case report describes the application of an innovative family-based diagnostic method (BOAM [16,17]) in a family with a child having problems with the self-regulation of emotions and behaviour, who will be introduced further in section 2 (Detailed case Description).”
Comment 12: (Lines 466-467): The discussion on autism heredity is somewhat implicit. A more explicit discussion in the main text or discussion section would improve clarity.
Response 12: Thank you for this idea. We now touched upon this issue by mentioning the heritability of autism in the text were we write on the daughter receiving her classification, and the parents feeling safe enough to reflect on themselves 26: “Also, their perception of a standard psychological assessment changed, which resulted in a DSM-5 classification for their daughter after finishing the BOAM-trajectory, namely autism spectrum disorder. The father also recognized the BOAM-diagnosis of overdemanding-related ordering problems in himself, and said he recognized some autistic traits in himself. This is not surprising, given the heritability of autism [51]. The BOAM-models helped the father giving words to his difficulties, which was important both for himself, and for the mother, and the way they worked together as parents. The guiding principle of the equal collaboration between the parents and the therapist probably also played a role in this case and possibly had a positive effect on the parents, who felt in control of the process, which created a safe space for self-reflection, gaining new perspectives and making new choices.”
Comment 13: The discussion section is too brief given the breadth of potential topics. As previously mentioned, several aspects could be explored in greater depth, such as comparisons with other tools and methodologies. For instance, Asperger emphasized the importance of free conversation with autistic children in contrast to standardized tools like ADOS or ADI-R. There is also a lack of references in this section. Expanding the discussion with relevant citations would add academic rigor. The conclusion is also too concise. It would be beneficial to include a discussion of the study’s limitations, future research perspectives, and potential improvements in methodology.
Response 13: Thank you for all your suggestions. We have significantly extended the discussion (page 26-29), and added references to it. Some additions were already clarified in response to earlier comments, but we have also added the following text to the Discussion:
“The influence of a therapist is very limited compared to the influence parents can exert on the psychological problems of a child or young person. After all, they are the attachment figures, role models as well as daily educators. That is why the BOAM method standardises family diagnostics and family treatment for all children and adolescents with (all sorts of) psychological problems. The method focuses on strengthening the parents' parenting skills in relation to the specific mental health problems, as well as the family dynamics between them. As a result of a BOAM-trajectory, a family diagnosis also emerges. This leads to a tailor-made parenting approach with which the parents can further address the psychological problems of their child or adolescent after treatment. Should follow-up assistance still be needed after BOAM self-diagnosis, this often consists of parental guidance in implementing this parenting approach and/or follow-up family counselling. Only if it is unavoidable and serves a clear purpose will additional individual treatment be provided for the child or adolescent. The parents of this case report also expressed the wish that their daughter would not receive individual treatment.”
“Meta-analyses also show that psychoeducation can be an effective tool in reducing children’s problematic behaviour in different populations of children, such as children with anxiety or attention deficit […, …].”
“The current case study gives initial support for this theory by showing that after focusing above mentioned factors, executive functioning improved and externaliziong behaviour decreased. Although prior research has not looked at ‘ordering problems’, evidence is growing for the importance of sensory processing difficulties in developing mental health problems […]. Also, executive functioning problems has been shown to be associated with psychopathology in youth [39,40].”
“With the help of the models, the parents made their daughter’s anger and aggression less unpredictable for them. Also, they gained insight on how to influence their daughters problems, which made them more manageable, and decreased their feeling of powerless.”
“This study has several strengths and limitations. A strength is that the case description was detailed, and many of the BOAM models were provided, making it possible for mental health professionals to use aspects of the BOAM-method. Another strength is that for some measures, we a waitlist assessment was available. The fact that there was no clinically significant improvement between the waitlist assessment and the pretest, suggests that the improvement that was found between pretest and posttest may be attributed to the BOAM trajectory and not to non-specific factors such as the passing of time. A limitation of this study is the fact that because it is a case report, it is not possible to draw conclusions on the general effectiveness of a BOAM trajectory, or on the generalizability of the results. Furthermore, the scientific value of this case report is limited because of the design that was used. A methodological improvement would have been to use a single case experimental design with multiple measurements per study period [61]. Another limitation was the fact that questionnaire data was not available for all measurement points and for both parents, and no other informants were included. “
Comment 14: I'm also not sure about the issue of conflicts of interest. Did the researchers create this tool? If so, this point should be presented and discussed.
Response 14: Yes, one of the authors indeed created the method. In the Declaration section, we declared that the second author (DT) is writing a book on BOAM with the support of two of the other authors. To be more clear, we added the fact that she is also the one who created the method in the Declaration section on page 30. Also, we added the following sentence to the Introduction on page 2: “The BOAM-theory and models were developed by author DT, while she was working with families with children with complex problems.” I think in this way we present this fact better than we did in the last version of the manuscript. You also suggested that we should discuss this issue, and we are a bit confused by this suggestion. Until now, we have been used to declare it in the declaration section but not mention it in the main text. Although we are willing to do so, we are also not sure what the main and secondary messages of this discussion should be. Maybe the editors can let us know how we should deal with this issue?
Comment 15: Once again, I would like to commend you on your work, as this case report makes an important contribution to the field. I look forward to seeing how the manuscript evolves with these refinements.
Response 15: Thank you very much for your encouragements to improve our work!